# Water Food Energy Nexus with Changing Agricultural Scenarios in India during recent Decades

Beas Barik[1], Subimal Ghosh[1, 2, *], A Saheer Sahana[1], Amey Pathak[1], Muddu Sekhar[3]

[1]Department of Civil Engineering, Indian Institute of Technology Bombay, Mumbai – 400 076, India

[2]Interdisciplinary Program in Climate Studies, Indian Institute of Technology Bombay, Mumbai – 400 076, India

[3]Department of Civil Engineering, Indian Institute of Science, Bangalore – 560 012, India

*Correspondence to*: Subimal Ghosh (subimal@civil.iitb.ac.in)

**Abstract**. Meeting the growing water and food demands in a densely populated country like India is a major challenge. It requires an extensive investigation into the changing patterns of the checks and balances behind the maintenance of food security at the expense of depleting groundwater, along with high energy consumption. Here we present a comprehensive set of analyses which assess the present status of the water-food-energy nexus in India, along with its changing pattern, in the last few decades. We find that with the growth of population and consequent increase in the food demands, the food production has also increased and this has been made possible with the intensification of irrigation. However, during the recent decade (after 1996), the increase in food production has not been sufficient to meet its growing demands precipitating a decline in the per-capita food availability. We also find a statistically significant declining trend of groundwater storage in India during the last decade, as derived from the Gravity Recovery and Climate Experiment (GRACE) satellite datasets. Regional studies reveal contrasting trends between Northern and West-Central India. North-Western India and the Middle Ganga Basin shows a decrease in the groundwater storage as opposed to an increasing storage over West-Central India. Comparison with well data reveals that the highest consistency of GRACE derived storage data with available well measurements, are in the Middle-Ganga basin. After analysing the data for the last two decades, we further showcase that after a drought, the groundwater storage drops but is unable to recover to its original condition even after good monsoon years. The groundwater storage reveals a very strong negative correlation with the electricity consumption for agricultural usage, which may also be considered as a proxy for groundwater pumped for irrigation in a region. The electricity usage for agricultural purposes has an increasing trend and interestingly, it does not have any correlation with the monsoon rainfall as computed with the original or de-trended variables. This reveals an important finding that the irrigation has been intensified irrespective of rainfall. This also resulted in a decreasing correlation between the food production and monsoon rainfall revealing the increasing dependency of agricultural activities on irrigation. We conclude that irrigation has now become essential for agriculture to meet the food demand; however, it should be judiciously regulated and controlled, based on the water availability from monsoon rainfall, specifically after the drought years, as it is essential to recover from the deficits suffered previously.

## 1 Introduction

The global population continues to rise steeply bringing forth an accordant increase of significant proportions in the demand for water-food-energy (Leck et al., 2015; Rasul and Sharma, 2015). However, natural causes along with anthropogenic factors have accelerated the rate of climate change. Consequent repercussions like the melting of glaciers and sea level rise, an increasing frequency and intensity of devastation by meteorologically related phenomena like droughts, flood, heat waves, cyclones and overall changes in weather patterns (Easterling et al., 2000; IPCC, 2012; Singh et al., 2014). The impacts of such events are long lasting, especially in the developing countries. Here, not only the time taken to recover from the losses in livelihoods incurred is long due the greater social and economic vulnerability, but the initial and continual loss of life is also huge during and in the aftermath of the event due to scarce access to water and food. The United Nations has defined food security and water security as , "food security is the condition in which all people, at all times, have physical, social and economic access to sufficient safe and nutritious food that meets their dietary needs and food preferences for an active and healthy life" and "water security is defined as the capacity of a population to safeguard sustainable access to adequate quantities of acceptable quality water for sustaining livelihoods, human well-being, and socio-economic development, for ensuring protection against water-borne pollution and water-related disasters, and for preserving ecosystems in a climate of peace and political stability" respectively. Similarly, the International Energy Agency (IEA) defines energy security as "the uninterrupted availability of energy sources at an affordable price". Hence, the nexus of water-food-energy has always been very closely interlinked to the sustenance and development of societies. Thus, maintaining the security of all the components of this tri-facet outlook is of immense importance both from the economic and social point of view (Devineni and Perveen, 2012; Wyrwoll, 2012; WWAP (United Nations World Water Assessment Programme, 2015)).

Water is required for agricultural produce, energy is required to pump the water from various sources, and again water is used for hydropower to generate electricity(energy). Hence, a vicious circle of water-food-energy is created where one imbalance leads to the deterioration of all the components. However, the global community has addressed each of the components separately but rarely have they addressed the problems that the three face together because of their close association (FAO, 2014; Endo et al., 2015; Rasul and Sharma, 2015).

India is the second most populous country in the world with an estimated total population of 1.2 billion (Census, 2011). Although, for India, the economic dependence in terms of contribution to the Gross Domestic Product (GDP) has reduced largely from the agricultural sector, with increasing emphasis on the industrial and service sectors (Eichengreen and Gupta, 2011; CSO, 2014; Patel et al., 2015); yet to maintain the food security, agriculture still has a major role to play. Though urbanisation is rampant throughout India, the percentage of rural population (68.84 percent) is still far greater than the urban population (31.16 percent) (Census, 2011). This stark difference brings up the underlying truth of agricultural dependency for food grains. The dwindling availability of fresh water poses a threat to food security in terms of reduced agricultural productivity and increased energy consumption for groundwater abstraction (Scott and Sharma, 2009) resulting in multiple complex challenges in terms of water security, food security and energy security along with the changing climate (Mukherji, 2007; Bazilian et al., 2011).

India has always been cited as a monsoon country where the months of June-July-August-September (JJAS) replenish the surface and groundwater stock (CGWB, 2014). These months also coincide with the highest agricultural activities of tilling, sowing and harvesting. 55 perecnt of the net sown area of the country is rain-fed, making a large section of the farmer dependent upon the monsoons for a good yield (NRAA, 2012). Thus, a good monsoon year is synonymous with high food grain production. However, recent studies have shown that there has been a weakening trend of the Indian Summer Monsoon (ISM) attributed to the warming of Western Indian Ocean (Roxy et al., 2015), high emission of aerosol in North India (Bollasina et al., 2011) and large-scale land use land cover change (Paul et al., 2016); but with an increase in the extreme rainfall events (Sinha et al., 2006; Goswami et al., 2006; Rajeevan et al., 2008; Paul et al., 2016). Another study suggests that the number of droughts and their severity will keep on increasing (Singh et al., 2014; Mallya et al., 2016; Salvi and Ghosh, 2016) in future. Hence, the belief that good monsoon years in future can improve the water crisis in the severely drought hit regions, is now a myth. Various other studies using the GRACE (Gravity Recovery and Climate Experiment) satellite datasets have also shown a decline in the snow fall in the Himalayan region with decreasing snow melt from the resident glaciers. During summer, these snow packs contribute as the fresh water sources for the major rivers, but presently this proportion has declined, leading to a negative storage trend of the total water storage (TWS) (Moiwo et al., 2011; Wu et. al., 2015). Such climatic variability is not favourable for a good agricultural yield neither in the immediately adjacent river basins nor in further downstream areas, which are very critically dependent upon this upstream supply for water and also for hydro-electricity generation (Rasul, 2014).

Along with climate change, human intervention has deteriorated India's water crisis further. Green Revolution has been blamed for falling groundwater levels due to over-exploitation of groundwater storage through uncontrolled pumping and energy made available at a subsidized rate (Mukherji and Shah, 2005; Badiani et al., 2012; Fishman et al., 2015; Zaveri et al., 2016). The Central Ground Water Board (CGWB), India which is responsible for monitoring the well depths, provides observation four times in a year; which made it impossible for having a continuous set of observations to analyse the fluctuations. However, with the launch of the GRACE satellite in 2002, a very important hydrological dataset, Total Water Storage (TWS), became available and the findings derived from it are of utmost importance. The first study done for North-Western India from August 2002 to October 2008, showed that the groundwater storage had declined without the effect of any stressor (normal rainfall conditions had prevailed in the interim), and this decline was thus attributed to the unmonitored abstraction of groundwater for irrigation and domestic purposes (Rodell et al., 2009). Another study in approximately the same area and of an over-lapping time-period also showed a decline, but the depletion rate was more pronounced during the first five years of study (2003 to 2007). Although, during the normal and excess rainfall years the groundwater storage did show an improvement, but overall, the trend remained negative (Chen et al., 2014). Similar analysis has been carried out for the Indo-Gangetic belt, a region with one of the highest agricultural and irrigation activities. Tiwari et al. (2009) have reported that in this region as well, groundwater depletion was prevalent. The main cause of such depletion is the over use or exploitative abstraction of groundwater. Other studies have also been carried out for the Indo-Gangetic Plain, with a few encompassing the Indus and the Brahmaputra-Meghna Basin as well. The outcome of all these studies, and their amalgamation, shows a decline in the groundwater storage levels, especially till the year 2009, and thereafter, an improvement or stability in the storage (Dasgupta et al., 2014; Prakash et al., 2014; Khandu et al., 2016; Yi et. al., 2016).

Such revelations are shocking because no longer can a drought be attributed simply to meteorological failure but such creeping hazards are now also dependent on how the water is used, especially when it comes to the usage of groundwater. However, no such studies have been conducted where the major factors have been looked in to causing such water crisis (i.e. depleting groundwater) based upon the available observed data. Literature suggests that anthropogenic activities to be the major factor for depletion of groundwater (Rodell et al., 2009; Tiwari et al., 2009; Dasgupta et al., 2014; Prakash et al., 2014; Khandu et al., 2016; Yi et. al., 2016), but a detailed tri-faceted analysis is still missing for India. Figure 1 shows the current scenario of the water-food-energy nexus along with climate change. A weakening trend of monsoon (Roxy et al., 2015) has reduced the amount of available water in India. Along with it a rise in temperature results in an increase in evapo-transpiration with an increased agricultural water demand. This water demand further increases due to agricultural intensification that is required to meet the growing food demand with population growth. Hence, with weakening monsoon and agricultural intensification irrigation, in terms of groundwater pumping becomes the major source of agricultural water demand. Often such use of irrigation becomes indiscriminate without monitoring of soil moisture. Although pumping of groundwater has multiple uses (domestic and industrial) but the major use (91 percent) is in terms of irrigation (CGWB, 2014). Thus, increased groundwater pumping results into depletion of groundwater and hence, more energy is required for irrigation in the subsequent years. Increased cost of irrigation affects the crop prices and that in turn makes food unaffordable to a fraction of population. This has resulted in the formation of a worsening scenario where, to maintain the food security groundwater and energy is over exploited.

Here we have tried to address the factors causing the accelerated groundwater depletion and its implication on food and energy security. Thus, the objective of this study for entire India is to not only evaluate the present status of groundwater storage changes but to also relate this to the varying amount of precipitation during the normal, excess and deficient monsoon (Supplementary Fig. S1) years. We have also examined how this relationship tends to affect the amount of groundwater usage, energy consumption and food productivity inter-annually. Further, sub-regions within India have been studied based upon their importance and contribution to the overall agricultural productivity. These sub-regional studies provide an incitement for developing better water management and usage practices, which has become the need of the hour for sustainable development.

**2 Study Area**

The Indian sub-continent lies roughly between 6 º N – 37 º N latitudes and 68 º E – 98 º E longitudes, covering an area of 3.287 million km². It has roughly four major seasons, pre-monsoon (March to May), monsoon (June-September), post-monsoon (October-December) and winter (January to February). Geo-physiographic features for the country varies from one part to another, as does the weather. The extreme western part of the country (comprising mainly of the state of Rajasthan with rainfall ranging from 15cms to 100cms), has an arid climate where as the eastern-most part (the Purvanchal Hills area with rainfall greater than 200cms), is one of the wettest regions (Ministry of Statistics, 2016) of the world. Such large variations result in different agricultural practices, crop planting strategies and schemes and accordingly the water usage is determined and modulated.

Indian agricultural production is divided into food crops and non-food crops. The essential food crops comprise of cereals (rice, wheat, bajra, maize, millets) and pulses (tur/arhar, gram). These are considered as the staple food for nearly the entire country. The non-food crops refer to oil seeds, cotton, tobacco to name a few, these are

important from the perspective of the economy to generate revenue. Hence, it is important to maintain a balance between the two, with priority given to food crops. Supplementary TableS1 shows a decadal variation of the area under cultivation, production, and percentage area under irrigation for food grains and cash crops. Also, Supplementary Fig.2 brings out the disparity in production and percentage area of irrigation for food grains and cash crops (specifically, sugarcane) very clearly. However, the area under cultivation is largest for food grains, despite that sugarcane has higher productivity. Farmers in many water deficit regions, due to lack of awareness, still cultivate water-intensive crops, hence aggravating the situation further (Siegfried et al., 2010).

Northern India is endowed with many perennial rivers, namely the Ganga, the Indus, the Brahmaputra and their tributaries, which serve as the lifelines for the flourishing agriculture in this region. Central and Peninsular India also has large rivers like the Mahanadi, the Godavari and the Kaveri. Due to climate change the amount of monsoon rainfall received has reduced (Rajeevan et al., 2008; Paul et al., 2016; Ghosh et al., 2016), hence the various river-interlinking projects that have been thought of by the Government of India (GoI) would see a rise in water-conflicts across different states (Bandyopadhyay and Perveen, 2008), which makes the optimum use of water resources both pertinent and of paramount importance.

Table 1 shows the major food grain producing states and their percentage share to all India production. These states are among the major contributors to the total food grain production for the entire country from the northern and central part of India. It is important to note that these states face a crippling water-crisis (CGWB, 2013-14), either due to a bad monsoon or due to the over-use and misuse of water. They are also among the most densely populated states of India (Census 2011). Hence, the problems not only pertain to climate variability but also with the population (FAO, 2014). Thus, the following sub-regions have been selected based upon their contribution towards the total food-grain production for India, population density (increasing pressure of food security) and the stage of groundwater development (higher percentage of development implies that the consumption has exceeded recharge).

The three sub-regions studied here are, North-West India (NWI) (the states of Rajasthan, Punjab, Haryana and Delhi) covering an area of 437,739.14 km$^2$; Middle-Ganga Basin (MGB) (the states of Uttar Pradesh and Bihar) covering an area of 339,488.09 km$^2$ and West-Central India (WCI) (the states of Maharashtra and Goa) encompassing an area of 311,249.34 km$^2$.

NWI, especially the state of Punjab, was the face of Green Revolution in India in the 1960s and 1970s (Rodell et al., 2009; Sharma, 2009). Support from the State and Central Government, enabled farmers to incorporate High Yielding Variety (HYV) seeds into the agricultural process and the additional benefit of using subsidized electricity to pump groundwater resulted in a  markedly increase of productivity. The immediate result was more than satisfactory but the long-term effect has been hard hitting (Singh, 2000; Hira 2009; Siegfried et al., 2010). The soil has been drained of its nutrients as the HYV seeds are water intensive, for which a well-developed irrigation facility is must. Since, the existent canal irrigation network could not provide this much-needed water supply, groundwater was exploited indiscriminately.

The MGB is one of the most fertile regions of the country, enriched annually by the alluvial deposits brought down by the various tributaries of the Ganges system from the Himalayas. It has a very dense network of irrigation.

However, farmers are still dependent upon groundwater because the staple crop of this region is paddy (highly water-intensive). Every time, the monsoon is delayed or fails, groundwater becomes the sole source of irrigation.

The WCI region, comprising primarily of the state of Maharashtra, has been regularly hit by droughts, frequently claiming the lives of many farmers, especially in the Marathwada and Vidarbha regions (interior Maharashtra). Since the onset of 2009, which was declared as a severe drought year for entire India, a continuous series of water crisis have plagued this region every year. However, the drought here, of late, has been human-induced rather than a meteorological failure (Udmale et al., 2014). Despite failed monsoons, many parts of the state continued water-intensive agricultural practices and farmers were not made aware of its consequences. The production of sugar cane and the operative sugar factories in this region are among the biggest water consumers. Hence, the need for an extensive study is required not only for the country but also for specific regions, which makes analysing the water-food-energy network's security extremely crucial.

### 3 Data and Methods

### 3.1 Data related to Indian Agriculture and Rainfall

The Directorate of Economics and Statistics, Department of Agriculture, and Cooperation, GoI has long-term datasets for India starting from 1950-51 to the present (Dacnet,2014). Based upon data availability of all other variables, the study period spans from 1950-51 to 2013-14. Here, we have showed the Indian agricultural scenario at a glance, as to how it has evolved over the last few decades. The important datasets considered for this study are the total food grain production (rabi and kharif) in million tonnes; per capita net availability of food (per annum) in kilogram per year; percentage of area under irrigation; different sources of irrigation; electricity consumption for agricultural purposes (proxy for groundwater withdrawal); and actual expenditure incurred by the agricultural department. All these datasets have been further analysed for a better understanding of agricultural practices and the dependency on irrigation. Here, for the present analysis, we use gridded rainfall data from India Meteorological Department (IMD) (Pai et al., 2014) at spatial resolution of 25 km × 25 km for the time-period of April 2002 to December, 2013.

### 3.2 GRACE Data for TWS

The GRACE satellite was launched in 2002 as a joint mission between the US (NASA) and German space agencies (DLR) to map the Earth's gravity field. The causal factors for time-variable gravity field are oceanic and atmospheric circulation, and changes in water storage on and within land. To isolate the water storage changes, the former two factors are removed through modelling (Swenson and Wahr, 2002). In this study, monthly land mass anomalies of 1 degree spatial resolution, release 05 level-3 data, from the CSR (Centre for Space Research, University of Texas at Austin, US) from April, 2002 to March, 2014 has been used. The level-3 dataset has been used after an extensive processing of spherical harmonics truncated at a degree and order 60. Also, in addition, a de-striping filter to remove spatially correlated errors resulting in north-south data "stripes" and a 300 km Gaussian smoothing were applied (Swenson and Wahr, 2006). Sampling and processing results in attenuation of the signal and to restore it, a scaling grid has been provided based on Community Land Model v4.0, which needs to be multiplied with the unscaled GRACE data (Landerer and Swenson, 2012). This mass anomaly dataset has a

time-mean baseline from January, 2004 to December, 2009. Hence, to make the other datasets comparable to the GRACE mass anomalies, this time mean baseline needs to subtracted.

TWS is a summation of soil moisture (*SM*), groundwater (*GW*), surface water (*SW*), snow (*S*) and ice (*I*) as shown in the following equation:

$$TWS = SM + GW + SW + S + I \qquad (1)$$

The climatology of TWS is subtracted from their respective monthly values, then it is multiplied with the area of the study region to get the anomaly of total water storage in terms of volume, where the positive (negative) anomaly refers to excess (deficit) storage. A detailed description about the estimation of TWS volume can be found in Thomas et al., 2014. Supplementary TableS2 gives an exhaustive list of scientific literatures using GRACE and GRACE derived groundwater studies specifically for the Indian sub-continents and its various sub-regions.

**3.3 Output of Global Land Data Assimilation System (GLDAS) and Estimation of Groundwater Storage Change**

Studies on GRACE data have isolated each of the TWS components using either observed data or modelled data to have an estimation of groundwater storage as shown in Eq. (2)

$$GW = TWS - (SM + SW + S + I) \qquad (2)$$

Here, the modelled auxiliary output from the Global Land Data Assimilation System (GLDAS) has been used to isolate each of the components from the TWS (Rodell et al., 2004), due to a dearth of observed data. The GLDAS has outputs from four models, Variable Infiltration Capacity (VIC), NOAH, MOSAIC and Community Land Model (CLM). Monthly 1 degree data has been used from April, 2002 to March, 2014 and an average of all the four model outputs has been considered in our analysis. The spread across the models has been considered as the estimate of uncertainty associated with this method. GLDAS outputs have been used for the estimation of groundwater from GRACE TWS over a decade in various parts of the world (Rodell and Famiglietti, 2001; Tapley et al., 2004; Andersen et al., 2005; Tiwari et al., 2009; Rodell et al., 2009; Taylor et al., 2013; Richey et al., 2015). Although surface water storage also plays a major role, due to the unavailability of data, it could not be incorporated (Rodell et al., 2007; Rodell et al.,2009). This is one of the limitation of the present study. The *S* and *I* component is zero for the regions selected and hence neglected. Finally, the following Eq. (3) has been used to calculate groundwater

$$GW = TWS - SM \qquad (3)$$

The *SM* is obtained from the GLDAS models to compute the groundwater storage (*GW*).

**3.4 Validation of satellite derived groundwater data**

GRACE data has shown a good agreement with in situ well depths for seasonal studies as well as a comparison with the TWS in studies conducted in Illinois (Swenson et al., 2006; Yeh et al., 2006), the Mississippi River Basin (Rodell et al., 2007), the High Plain Aquifer, Central US (Strassberg et al., 2007; Strassberg et al., 2009), in the

Hai River Basin, North China (Moiwo et al., 2009; Feng et al., 2013), also in the deltaic region of Bangladesh (Shamsudduha et al., 2012). The number of studies on validation of GRACE data for Indian regions is limited. One of them is for the Gangetic Plain where the investigators have validated the Δ TWS with a summation Δ SM and observed Δ GW; and have found a good agreement (Dasgupta et al., 2014). The other study has been conducted over the Gangetic Plain and a portion of North-West India (State of Punjab) wherein a total of 2800 and 250 wells for Gangetic Plain and Punjab respectively were analysed. (Panda and Wahr, 2016). The months of May and November show a high correlation of 0.92 and 0.84 respectively between the GRACE derived groundwater and the observed groundwater depths.

The CGWB is the responsible body for monitoring well depths and groundwater quality across India. Its network of wells comprises of exploratory wells, observations wells, piezometers and slim wells. Well depth readings for approximately 22,964 groundwater observation wells are measured by the CGWB, four times in a year (January, May, August and November), and this data is available from 1996 to 2014. However, the drawback of these observations is that they are not continuous, a problem further compounded by a number of missing observations for several of these wells for certain interim durations. The entire network too does not have the same historical lineage, having been setup at different times and phases. This well depth data was downloaded from a public domain. After filtering all the available well dataset for a continuous observation, a total of 743 wells for the MGB, 626 wells for NWI and 860 wells for WCI have been analysed for the validation of GRACE data in the present study.

**4 Results and Discussions**

Section 4.1 present results of the current scenario of agriculture in India, and the existent relations between groundwater, food and energy security. Section 4.2 present the validation results for GRACE derived groundwater and CGWB in situ well depth, followed by Sect.4.3 where the food-water-energy nexus for India has been described, and each of the component has been dealt with separately. Finally, Sect. 4.4 presents the nexus for the three sub-regions separately.

**4.1 Overview of Indian Groundwater, Agriculture and Energy Scenario**

The initial analysis of the various datasets that are available on official government websites reveals that although the food production has increased considerably but not to an extent to support the ever-increasing population. Figure 2a shows that during 1950-51 total food production was about 50.8 million tonnes which has increased to approximately 252.22 million tonnes in 2016 (based on the 4[th] advanced estimate by the Ministry of Agriculture and Farmer's Welfare, GoI), which seems to be at par with the rise in population.  During the 1960s with the advent of Green Revolution the initial scare of wide-spread famine was wiped out and food security was maintained. However, the net per capita availability of food shows a decline post 1996 as shown in Fig. 2b, which brings forth the clear picture of a decline in the per capita food production, but a steady increase in the rate of population growth. Food grain production over the years does not always show a steady rise, as it is dependent upon the climatic condition and inputs available (irrigation, fertilisers, good variety of seeds). The food production

showed a clear drop coinciding with each drought year, this drop was very pronounced for the year 2002 (major drought/ deficit monsoon year). However, for the drought during 2009, the dip was not large.

During the drought year of 2002-03, the food production was reduced by nearly 38 million tonnes as compared to the average production, but during the drought year of 2009-10, the reduction was only by 16 million tonnes. This comes as a surprise as to what caused the restrained fall of total food grain production, despite a severe drought year. Percentage area under irrigation has risen considerably from 18 percent in 1950 to nearly 50 percent in 2011-12. This increase in percentage of irrigated land has a high correlation with the total food grain production (R=0.99), the detrended correlation between the two is 0.46. This is probably the reason behind a smaller drop in food production during the severe drought year 2009-10, as compared to other drought year 2002-03. Figure 2f shows the different sources of irrigation and the contribution of groundwater stands out clearly. CGWB (2014) report states that nearly 91 percent of the groundwater abstracted is used for irrigation alone. Sufficient groundwater for irrigation is made available to the farmers, along with energy with low or no tariff, no matter to what depth the groundwater may fall (Mukherji and Shah, 2005; Badiani et al., 2012; Fishman et al., 2015; Zaveri et al., 2016). Such schemes were in place during Green Revolution and there have been no major changes in them till date (Mukherji and Shah, 2005; Badiani et al., 2012; Fishman et al., 2015; Zaveri et al., 2016). The brunt of the availability of subsidised energy is borne by the Government of India (GoI) (Kumar, 2005; Rattan and Biswas, 2014). This causal relation is clearly observed through the positive correlation (R=0.89) between the actual expenditure by the department of agriculture and co-operation and the electricity used for agricultural purposes (Fig. 2g and h). This implies that with the increase in electricity consumption (pumping of groundwater), the expenditure incurred for agriculture also increases. Hence, a grave situation has been created where food security is maintained at the expense of groundwater and energy security (Rosegrant and Cline, 2003).

Since, groundwater is the major source of irrigation a detailed study for it is required at a national and sub-regional level, which has been discussed in the next sections. Before performing the analysis, the GRACE derived groundwater storage is validated against the observed well data.

## 4.2 Validation of GRACE derived groundwater data

CGWB well depth data has been compared with GRACE derived groundwater to validate it. Figure 3a shows the three sub-regions considered for our region-specific study. Figure 3b, d and f show the groundwater depths measured in meters below ground level (mbgl), where zero refers to the groundwater at the surface (opening of the observation well) denoting no change in the water table. A time series of the four months have been plotted from January,1996 to January, 2014.

Figure 3c, e and g show the correlation between GRACE derived groundwater and the groundwater depths measured by CGWB. The four different coloured dots in the correlation plots denotes the four months of January, May (pre-monsoon), August (monsoon) and November during which data is collected by CGWB.

Figure 3b shows a negative trend (not significant) of the groundwater depth for MGB. In the monsoon month of August, a correlation of -0.17 is observed between the CGWB measured groundwater depths and GRACE derived groundwater, similarly correlation for the months of May, November and January have been calculated which are

0.36, -0.15 and 0.37 respectively. Overall a moderate positive correlation (R = 0.37) exist between the two (Fig. 3c), hence the GRACE derived groundwater can be used for further analysis.

Figure 3d is the groundwater depth for WCI and it shows an increasing trend (statistically not significant), similar to the GRACE derived groundwater trend (in Sect. 4.4.3) and they also have a positive correlation (R = 0.46) as seen in Fig. 3e. The individual correlation for the months of January, August, May and November are 0.35, 0.16, 0.33 and 0.13 respectively. The low correlation might be due to the lack of continuous observation from CGWB for all the months across the years.

Figure 3f shows the groundwater depth for NWI and it does not show any trend neither it has a correlation with the GRACE derived groundwater (R = 0.05) in Fig. 3e. This no correlation may be attributed to the individual poor correlations of the months of January (R = -0.55), May (R = 0.3), August (R = -0.13) and November (R = -0.06). Especially, for the month of January which has a highest negative correlation, months of August and November also are negatively correlated. Only the well data for the month of May seems to have a moderately good agreement with GRACE.

We observe that for MGB and WCI, for the months January and May, the well data and GRACE data have a moderate positive correlation, however when it comes to the months August and November they are negatively correlated. This is probably because during monsoon and post-monsoon surface water storage has a significant value and we neglect the same in deriving GW from Eq. (2). Further to this, the well data available from CGWB are not continuous and the sample size is also low. Under such situation, with spatially and temporally discontinuous ground observations, a high correlation may not be expected. However, except one region (NWI), we have got statistically significant correlation between GRACE derived storage and well depth data (p-value < 0.05) and this justifies the use of GRACE for the present study to estimate groundwater storage. Use of satellite derive surface water storage in derivation of GW from GRACE may be considered as a potential area of future research.

**4.3 Status of Water-Food-Energy Nexus in India**

In this section the three components of the nexus have been dealt with separately and finally the association between the three is established.

**4.3.1. GRACE derived groundwater and TWS**

GRACE and GLDAS data has been used to compute the variation in TWS (Fig. 4a) and changes in groundwater (Fig. 4c).

Figure 4a depicts the variation in TWS with the solid blue line, the dashed line shows the negative trend of the TWS and the shaded green portion shows the deficit (negative) and or excess (positive) in TWS volume. In the year 2002 it shows a slight deficit of -16.7 $km^3$ in August and -53.7 $km^3$ in September, however from 2003 onwards to 2008 there has been no more deficit in TWS. This is because, after a failed monsoon in 2002, the consecutive years of 2003 and 2004 had very good rainfall, which replenishes the TWS. However, with the onset of the 2009 drought (severe) the excess TWS is depleted and it shows a large deficit in 2009 and 2010. The maximum deficit was in the four months of monsoon, due to two reasons. Firstly, inadequate rainfall could not

replenish the TWS and secondly due to lack of surface water, groundwater must have been utilised for all anthropogenic activities resulting in a further deficit. Although, the deficit reduced largely in 2011, but again during 2012, India did not receive sufficient rainfall because of which the deficit further continues. It should be noted that the agricultural productivity was not affected severely during 2009 due to the use of groundwater storage and that resulted in a significant decline of it as observed in Fig. 4a and c.  Hence, here we see a clear example of a meteorological drought resulting in hydrological and socio-economic drought. Hence, the quantification of the TWS could be considered as an indicator for hydrological drought monitoring (Houborg et al., 2012; Yang et al., 2014). TWS is observed to have a significant (p-value: 0.02) decreasing trend.

Changes in groundwater storage (derived from GRACE) have been computed and it also shows a significant negative trend (p-value:$2.021\times10^{-13}$). Figure 4c shows the GRACE-GLDAS derived groundwater storage for India. The red line represents the multi-model average, which has a significant negative trend (p-value:$2.02\times10^{-13}$). VIC and NOAH has the least deviation from the mean, whereas CLM and MOSAIC has a larger deviation resulting in an increased uncertainty. Here, we find that similar to TWS, the fall in groundwater storage in 2009 has not been able to recover as the severe drought behaves like a shock to a balanced system. To get the system back to equilibrium the groundwater needs to reach its previous normal state. TWS and the groundwater storage has moderate correlation with the All India Monsoon Rainfall (AIMR) with the values 0.37 and 0.23, respectively Fig. 4b and d. Since groundwater is considered to be the major component of TWS, fluctuations in groundwater is replicated in TWS as well. The falling levels of TWS and groundwater, takes a long time to be replenished by rainfall through recharge.

In Fig. 4e and f GRACE derived groundwater, the average soil moisture from GLDAS modelled output and AIMR has been plotted (all three-time series have been normalised and made comparable to GRACE as mentioned in Sect. 3.2). A very important observation of the over-exploitation of groundwater has been brought out here. Since the year 2002 was a drought year, the soil moisture is depleted significantly by the time monsoon arrives in 2003.The arrival of a good monsoon recharges both the groundwater and soil moisture. Till the end of 2007, anomalies of groundwater and soil moisture over-lap each other, but a difference between the two begins from 2008 onwards and it further gets amplified in 2009 and continues for the later years. The year 2008 clearly shows a fall in GRACE derived groundwater and the soil moisture being not depleted to that extent, similar, trend has followed till 2014. The non-depletion of soil moisture during 2008-09 is due to excessive groundwater pumping and application of irrigation. Also, the correlation between AIMR and GLDAS soil moisture is observed to be 0.49, which is a moderately good correlation and it can be stated that rainfall also contributes to recharging of the soil moisture considerably. Hence, our belief is that soil moisture is adequate to support crop-water requirements with an optimized application of irrigation (if needed).

 Hence, our finding confirms that the misuse and over use of groundwater is prevalent since no proper policy or regulation is in place which will be able to monitor the withdrawal of groundwater and effectively use the available soil moisture (Siegfried et al., 2010; Badiani et al., 2012; Fishman et al., 2015). Hence, the current scenario for India as far as groundwater is concerned is not a satisfactory one as its rapid decline brings about the question of food, water and energy security.

### 4.3.2. Groundwater and Energy Consumption

Electricity consumption for agricultural purposes has been used as a proxy for energy used in groundwater withdrawal, since a major portion of energy is used in pumping of groundwater (Shah et al., 2004; Planning Commission Annual Report 2013-14). The annual GRACE derived groundwater anomaly is analysed with the electricity consumption data for a common period of 2002-2003 to 2011-2012 (following the data availability of power consumption based upon the financial year for India starting from April of a year and ending in March of the consecutive year). As mentioned in the previous section, from 2008 onwards the spell of monsoon was weak, followed by 2009, which was a severe drought year. In Fig. 5a, we observe negative anomaly of groundwater storage from 2008-09 to 2011-12. The consumption of electricity has increased during the same period. GRACE derived groundwater storage change has a negative correlation, Fig. 5b (R = -0.86) with the consumption of electricity. This signifies that the falling depth of groundwater results in greater amount of energy consumption to pump it for irrigation.

A similar analysis has been conducted for the progressive relationship between the observed groundwater table depth and electricity consumption for pumping, i.e., there exist a correlation between the annual change in groundwater level (year 2-year1) and the electricity used for pumping in year2. We plot the same for the three regions, MGB (Supplementary Fig. 3(a)), NWI (Supplementary Fig. 3(b)) and WCI (Supplementary Fig. 3(c)). The duration of the analysis is from 1999 to 2011.

We find statistically significant positive correlation between groundwater level drop and electricity consumption for MGB. However, statistically significant correlation does not exist for NWI and WCI. For WCI, this is expected and it is consistent with overall increase in groundwater level that possibly attributes to judicious use of groundwater. However, a careful investigation for NWI reveals that the correlation value is dominated by two outliers (marked in red in Supplementary Fig.3(b)) of changes in groundwater table depth. After removing the outliers, we obtain a very high statistically significant correlation as presented in Supplementary Fig.4.

To test the hypothesis that normal or excess monsoon years should have a lesser energy consumption (due to lesser groundwater pumping), we present a scatter plot (Fig. 5d) between AIMR and electricity consumption for agricultural purposes. Energy use in agriculture has two major usages: pumping groundwater for irrigation (electricity) (Scott and Shah, 2004; Birner et al., 2007; Kumar et al., 2013) and mechanisation (diesel) due to the use of tractors (Jha et al., 2012). Nearly 83 percent of the available water resources is used for agricultural activity, wherein 91 percent of the groundwater abstracted is used for irrigation purposes (CGWB, 2014). The agricultural electricity tariffs in India have been kept low, keeping in mind the poor economic status of farmers to facilitate groundwater pumping (Badiani et al., 2012). Due to low tariff, farmers have considered groundwater as a continuous affordable source of freshwater leading to an uncontrolled use of the same even if there is a good amount of monsoon rainfall in a specific year. The irrigation is no longer agricultural demand driven but rather dependent on the availability of electricity at a lower tariff. The scatter plot between AIMR and electricity consumption for agricultural purposes represent the same with no statistically significant correlation between them. Further to this, the irrigation practice in India is mostly flood irrigation that has very poor irrigation efficiency. This has significant implications in terms of not only agricultural water management policy but also on hydrological simulations and modelling studies. Traditionally, state of art land surface models does not consider irrigation and even if they consider irrigation, the water use is demand driven. The situation of

agricultural water use in India is far from the model assumptions and hence model driven studies often underestimate the agricultural water use and groundwater abstraction. Model derived groundwater abstraction shows high dependability on monsoon rainfall (Asoka et al., 2017); however, the same needs to be tested further with ground truth. Implications of uncontrolled flood irrigation in India have been reported by Devineni et al. (2013) and Fishman et al. (2015) and this needs to be further explored to understand and realistically simulate the water cycle of Indian subcontinent.

### 4.3.3. Groundwater-Food-Energy Nexus

A similar analysis has been conducted for food production and groundwater storage change. As the food production has increased there has been a decline in the storage of groundwater (Fig. 6a) and they have a high negative correlation (R= -0.73) as seen in scatter plot (Fig.6b). On the other hand, a positive correlation (R = 0.96) exists between the food production and electricity consumption (Fig. 6d). This implies the high dependency of food production and groundwater irrigation.

Energy usage in agriculture has two major usages: pumping groundwater for irrigation (electricity) (Scott and Shah, 2004; Birner et al., 2007; Kumar et al.,2013) and mechanisation (diesel) due to the use of tractors (Jha et al., 2012). Several studies have stated that the electricity for agricultural purposes is mainly used for irrigation (Scott and Shah, 2004; Birner et al., 2007; Kumar et al.,2013) because the farm mechanisation is dependent upon diesel (Jha et al., 2012). Hence, an assumption that has been considered for this analysis is that, the data for 'consumption of electricity for agricultural purposes' represent the electricity used for pumping groundwater.

In Figure 6c we observe that in the year 1957-58 the electricity consumption was 544.64 GWh along with a food production of 64.31 million tonnes. 20 years later the electricity consumption increased by nearly 20 times (10107.36 GWh) but the food production increased only by twice the previous (126.41 million tonnes). In year 1997-98 electricity consumption was 97195 GWh but still the food production lagged (203 million tonnes). In the recent year of 2011-12 food production increased to 259.32 million tonnes and electricity consumption was 140960 million tonnes. Hence, over the past 54 years (1957-58 to 2011-12) electricity consumption has increased by more than 250 folds whereas the food production has increased only by four folds. Thus, when actual observed values are considered we see that electricity consumption has increased in leaps and bounds but food production has failed to do so, which brings out the concern regarding food security. Thus, there exists a clear dis-balance between the three facets of the nexus. This indicates that the food security has been maintained at the cost of water and energy security.

Figure 6e shows an unexpected relation between AIMR and food production. After the year 1986-87 total food production has not been largely dependent upon the AIMR. In spite of a decline or increase in the monsoon (Supplementary Fig. S1) the total food production has remained more or less unaffected. In Fig. 6f a time series of correlation between food production and AIMR, for 30 years moving window, has been plotted and it shows a statistically significant (p-value:$8.12\times10^{-12}$) negative trend. This was the least expected change that has taken place over the last two decades. A detrended correlation of the two variables also show a similar significant (p-value:0.001) negative trend.

In this section, we have tried to bring out a complete picture of the nexus, where the groundwater has declined and the electricity consumption has increased, this is further supported by the high correlation between total food production and energy consumption again. However, the major issue which needs to be addressed soon, is the apparent declining dependency of food production on AIMR, because of which the usage of groundwater has

increased even during normal or excess monsoon year. Majority of the studies and reports have mentioned that the agricultural production and hence India's economy is dependent on monsoon (Kumar et al., 2005; Gadgil and Gadgil, 2007; Gornall et al., 2010) as large number of areas are rain fed, but that no longer hold true and needs a more detailed local investigation.

### 4.4 Status of Water-Food-Energy Nexus in Sub-Regions

Apart from an analysis at a national level, regional level studies are also very important, because every region has a different climate regime depending upon which farmers should develop their agricultural practices and policy makers should develop water and energy management schemes.

### 4.4.1. Middle-Ganga Basin (MGB)

This region is one of the most densely populated areas of India, encompassing the states of Uttar Pradesh and

Bihar. They have very high population density of 828 persons per $km^2$ and 1102 persons per $km^2$ respectively with a very high fraction of population being farmers. Similar to the study for all India (as described in Sect. 4.3), GRACE data has been used to analyse the TWS, changes in groundwater storage and their relation with rainfall in this study area. The TWS for MGB follows a pattern very much alike to that for all India. The deficit TWS sets in much earlier than the 2009 drought year. A small deficit takes place towards the end of 2007, but it is recovered

during the monsoon of 2008. However, the recovery from the 2009 deficit could not take place as the negative anomaly in water storage was very high. During the monsoon months, a mean deficit of 37.32 $km^3$ of TWS had set in, which further declined to 41.95 $km^3$ in 2010 (Supplementary Fig. S5a). TWS has a contribution from rainfall as seen from their positive correlation (R = 0.42) in Supplementary Fig. S5b howsoever it still shows a significant (p-value:$9.76\times10^{-7}$) declining trend. The groundwater storage has a lower correlation (R = 0.29) with

rainfall (Supplementary Fig. S5d). Supplementary Fig.S5c shows the GRACE-GLDAS derived groundwater storage for MGB. Similar, to the analysis for India, here the derived groundwater also shows a statistically significant negative trend (p-value: $2.58\times10^{-21}$). Deviations for the simulations from MOSAIC and CLM are higher than those from VIC and NOAH. Both, MOSAIC and CLM derived groundwater shows a higher deviation specifically around the year 2008-09.. This is probably due to the water intensive agricultural practices prevalent

even during the drought years. Here, we observe that in the absence of surface water, groundwater is extensively used to suffice the irrigation demands, and thus it is not given an opportunity to recharge. Even during the good monsoon year of 2007, a deficit was observed, and hence for a severe drought year a worse scenario is foreseeable.

The change in groundwater is also drastic after 2009, and it has not been allowed to be replenished to its original state back in 2003. Although it shows a negative trend till our study period of March, 2014, the only relief being

that the groundwater is stable and does not show further decline, rather there has been a slight improvement.

In this region-specific study as well, apart from the deficit of 2007 as mentioned previously, the soil moisture seems to be well maintained, and the pattern is followed through 2009 till 2014 (Supplementary Fig. S5e). The

soil moisture has a good positive correlation with rainfall (R= 0.46), implying that rainfall can recharge the soil moisture to a certain extent. Additionally, due to continuous abstraction of groundwater soil moisture is able to recover quickly from the negative anomaly in 2009 (Supplementary Fig. S5e). This results in a replenished soil moisture and depleted groundwater scenario.

As far as the energy consumption is concerned, it has increased with a depleting groundwater (Supplementary Fig. S6a and b) and has a high negative correlation (R = -0.81). As previously seen, the contribution of rainfall towards recharging of the groundwater is low due to severe anthropogenic activities, and this condition also holds true here where the electricity consumption for agricultural purposes keeps increasing without considering the real need based on monsoon rainfall (Supplementary Fig. S6c and d). This increased consumption of electricity is synonymous to excessive groundwater irrigation even in surplus years, which results in loss of groundwater and energy.

The nexus for the MGB also depicts the food security being sustained at the cost of groundwater and subsidised energy provided for pumping it from falling depths. Supplementary Figure S7 shows this same scenario of increased food productivity with increased electricity consumption, but with concomitant falling of groundwater levels. The concern here is not only with depleting groundwater but also contamination of it by arsenic or salt water intrusion as a recent study has pointed out (MacDonald et al., 2016; Fendorf and Benner, 2016).

### 4.4.2. North-West India (NWI)

This region has faced the tremendous impact of Green Revolution, initially as a blessing, but later considered to be harmful. Howsoever, not much has been done regarding it, the policy of subsidised energy to pump groundwater was started here and it has since continued (Mukherji and Shah, 2005). Since this region is in an arid zone, its dependency on groundwater for irrigation is high (Suhag, 2016).

The TWS here shows a continuous fluctuation from a deficit to excess, but after 2009 it has been markedly in a deficit state and it follows a declining trend (p-value:$4.17 \times 10^{-8}$). Initially, the TWS was at its peak in April, 2002 ($29.15$ km$^3$) (Supplementary Fig. S8a) till the beginning of 2004. However, due to an erratic rainfall pattern, the excess storage started declining with few deficits apparent in between in 2007 and 2008. The TWS storage deficit reached its nadir of $29.1$ km$^3$ in May, 2010, after which it regained rapidly in September, 2011 and had a deficit of $8.27$ km$^3$ (Supplementary Fig. S8a). This improvement cannot solely be attributed to the groundwater, since soil moisture is also a contributor to the TWS. Supplementary Fig.S8c shows the GRACE-GLDAS derived groundwater for NWI. Here as well the multi-model average line has a significant negative trend (p-value: $6.79 \times 10^{-29}$). The uncertainty is low compared to the other regions. 2011 was an excess rainfall year in this region owing to which the groundwater recovered considerably and the soil moisture was also replenished (Supplementary Fig.S8c and e respectively). Similar findings as in the other areas are observed, where the soil moisture is in excess but the groundwater is deficit. TWS and groundwater are not much influenced by the rainfall, but soil moisture has a positive correlation with it (R = 0.41) (Supplementary Fig.S8b, d and f). Over all the GRACE derived groundwater has a decreasing trend (p-value:$6.78 \times 10^{-29}$).

Energy consumption and groundwater depletion are negatively correlated (R = -0.9) denoting greater amount of subsidised energy consumption to pump groundwater from falling depths, making the irrigation sector the main

cause for such a situation (Supplementary Fig. S9a). However, the relationship between rainfall and electricity consumption shows a positive correlation (R = 0.45), denoting that in spite of there being a few good rainfall years, the dependency on groundwater does not decline. Groundwater is still pumped irrespective of a surplus monsoon year hence the higher consumption of energy. (Supplementary Fig. S9c). This also points to the fact that replenishment of groundwater does not occur, be it a normal/excess/deficit year ground ware is pumped indiscriminately resulting in loss of it and also energy.

The results for the food security aspect is similar to that for the MGB. Here as well, with increased food production the groundwater is depleted (R = -0.83) and the consumption of electricity for agricultural purposes increases with increased food production (R = 0.79) as shown in Supplementary Fig. S10.

The conclusion from the above analysis is that a proper water management policy needs to be put in place very soon which can impart knowledge to farmers as to the kind of crops (Morison et al.,2008; Fishman et al., 2015) which may be planted during drought years, or how the available soil moisture can be used, without over using groundwater for irrigation. There is a need to develop a proper irrigation scheduling considering the weather forecasts (Shah et al., 2016) and extended range forecasts (Sahai et al., 2013) of precipitation.

### 4.4.3. West-Central India (WCI)

WCI is comprised almost wholly by the states of Maharashtra and Goa. For the last few years in this region, the farmer mortality has increased due to a meteorological drought that started way back in 2002. This region is also sensitive to changes in precipitation patterns inter-annually. The TWS had a deficit from 2002 to early 2006, while the rainfall improved in the years of 2005 and 2006 resulting in an excess of TWS till the middle of 2008. Again, 2008 onwards, the rainfall was not sufficient and along with it, the 2009 drought resulted in a decline. From 2010, it showed an improvement (Supplementary Fig. S11a and e). In the time-period of our study the maximum deficit of storage was of 38.53 $km^3$ in August, 2002 whereas the maximum excess storage was 36.49 $km^3$ that reached in October, 2013 (Supplementary Fig. S11a). In comparison to the other two regions, the WCI showed an increasing trend during the post 2009 drought for TWS (p-value: $2.11\times10^{-7}$). Supplementary Fig.11c shows the GRACE-GLDAS derived groundwater for WCI. The multi-model average also has an increasing positive trend (p-value: $2.1\times10^{-7}$). MOSAIC derived groundwater has a higher deviation from multi-model average through the entire study period.

However, such a finding is contradictory to the scenario in Maharashtra, but we need to keep in mind that the minimum spatial resolution for GRACE is 150, 000 $km^2$, and any area less than this would give high errors. Hence, the study area is the entirety of Maharashtra and Goa and the findings give us an over view for the entire state.

The Ground Water Year Book published by CGWB reports a comparison between the August 2012 groundwater level and the decadal mean (August, 2002 to August 2011), which shows a rise in the groundwater level in Maharashtra. As far as the depletion trend of the groundwater goes, it is concentrated only in few areas of Nashik, Ahmednagar, Jalgaon, Solapur, Amravati, Sangli and Buldana (GSDA, 2014). These areas are far smaller than the GRACE footprint, and hence an analysis for each of the regions has not been done.

The overall consumption of electricity for agricultural purposes has a positive correlation (R =0.39) with the depletion of groundwater. This denotes that higher the availability of groundwater, more is the electricity

consumed and vice versa (Supplementary Fig. S12a and b). This aspect has also been reported by the GSDA(Groundwater Surveys and Development Agency, GoM), where there is a shift in agricultural practice; no longer water intensive cash crops (sugar cane, banana, grapes, oranges) are solely planted. Also with improved surface water irrigation, return flow and curbed usage of groundwater has resulted in its recharge. In many regions, such as in Nagpur, Pune, Akola and Kolhapur, such improvements have been reported by the GSDA, 2014. Similarly, for food production as well, the groundwater does not seem to make a significant contribution, they are both positively correlated (R = 0.33), referring to an increase in both (Supplementary Fig. S13a and b). This is only possible if there is another source of fresh water which is used for agriculture and it also replenishes the groundwater.

Thus, the state of Maharashtra puts forward an example to a balanced nexus of water-food-energy. However, the increase in mortality due to farmers' suicide shows the need of socio-hydrological analysis (Udmale et al., 2014; Pande and Savanije, 2016) which may be considered as the scope for future research.

**5. Summary and Conclusion**

This present study is the first of its kind for India, encompassing all the three major sectors of water, food and energy from the perspective of the agricultural sector. Based upon MoWR (Ministry of Water Resources, GoI) (2012), the per capita availability of fresh water has declined from 1816 cubic meters considering the population based on the 2001 census to 1545 cubic meters as per the 2011 census. Hence, the water stress is evident due to the continuous increase in demand by the increasing population (CWC, 2013). The total annual replenishable groundwater potential of our country is 431 billion cubic meter (BCM), Uttar Pradesh (17.5 percent) has the largest share, followed by Maharashtra (8.3 percent), Bihar (6.6 percent), Punjab (5.2 percent) and Rajasthan (2.8 percent). A study by Amarasinghe et al.(2004) found that high values of the Ground Water Abstraction Ratio (GWAR), which is defined as the ratio of groundwater withdrawals to groundwater availability and refers to the degree of development of groundwater resources in a region, coincided with 'unsustainable use of groundwater'. Surprisingly, the regions then marked out for following 'unsustainable use of groundwater' practices still have a similar situation at present. The three sub-regions studied here coincide with the finding of this study. MGB and NWI had the highest GWAR, ranging between 51 percent to 197 percent  and WCI varied between 25 percent to 50 percent. Our results show similar depleting trends for MGB and NWI, but for WCI there is an improvement. This depletion of groundwater is caused due to unsustainable use and agricultural practices which do not adhere to the region's water balance equation. Crops such as sugar cane and cotton are highly water intensive, more so since they are cash crops. Hence, they not only utilise excess water (groundwater) but they also do not contribute towards the food security. Cash crops, as the name suggests, are produced as they are more of revenue generating crops. Thus, the governing body needs to realise that on one hand it is supplying energy at subsidised rates which is leading to groundwater pumping but this goes towards cash crop production.

The threat to the security of the nexus might be further intensified as these three zones lie in region which are highly sensitive to climate change and large percentage of districts fall under the category of high vulnerability (Rao et al., 2016). Thus, to combat climate change and to ensure security for water-food-energy strict norms of

crop cultivation needs to be in place. Following are few measures which could be incorporated for judicious regulation and control:

- **Soil Moisture Monitoring and irrigation practice:** Uncontrolled use of groundwater for irrigation attributes to the lack of monitoring of soil moisture and hence the irrigation practices in India are not demand driven. Recent studies (Devineni et al., 2012; Devineni et al., 2013 and Fishman et al., 2015) show that the application of irrigation based on soil moisture condition may result into conversion of significant area from water stressed to water surplus. Further to this, the irrigation practice in India is largely flood irrigation, which has very low efficiency. Changing of irrigation type from flood to drip may reduce significant water wastage and improve irrigation efficiency.

- **Considering seasonal and extended range forecast for better water consumption:** In India, the agricultural water management models exists in theory, but they are seldom used in practice. An agricultural water allocation model at a fortnight scale or at a seasonal scale considering the improved seasonal (Saha et al., 2012) and extended range forecast (Sahai et al., 2013; Shah et al.,2016) may be useful and this needs to be further explored in near future.

- **Tariff determination based on seasonal prediction (varying volume of water usage):** Based upon seasonal prediction, a farmer would be made aware of a good/bad/normal monsoon year. Demands should be calculated based on seasonal prediction and accordingly the required amount of groundwater irrigation may be obtained considering possible margin of errors from hind-cast data. This amount may be considered as an upper limit for the famers to be used freely or at a subsidized tariff. If this upper limit is crossed, then they would not be allowed to enjoy free water and cheap electricity. This would majorly reduce the over-exploitation of groundwater.

- **Groundwater usage metering:** Electricity has been made available at a subsidised rate (almost free), increasing the accessibility of groundwater (Badiani et al.,2012; Fishman et al., 2015). If a volumetric tariff on groundwater (Shah et al., 2004) usage at a local level or at least at a district level is put in place it would also help in estimating the groundwater abstraction and accordingly management of the same.

- **Water allocation and water pricing:** Water allocation system needs to be put in place depending on the productivity and yield of crops (Huh and Lall, 2013). Water allocation and pricing should follow an optimal cropping policy (Devineni et al.,2012). Cultivation should be carried out in an area where it is best suited for a specific crop. Hence, reducing the pressure on the resources to produce the crop.

- **More production of food grains rather than cash crops:** Farmers have an inclination of producing cash crops than food grains as they are economically more lucrative. Food grains have a minimum price policy that is fixed by the government, hence profit maximisation is low. Moreover, most of the cash crops used are water intensive. Policy interventions are required to manage the same for a sustainable water and food security.

*Data availability* All data sources are mentioned within the text and also in the references.

*Author Contribution* SG and BB designed the problem and the solution algorithms. BB collected all the data. BB, SAS and AP processed the GRACE Data. BB performed the analysis. BB, SG and MS interpreted the results. BB and SG prepared the figures and wrote the manuscript. All the authors reviewed the manuscript.

*Competing interests* The authors have declared that no competing interests exist.

5 *Acknowledgements* The authors gratefully acknowledge the financial support provided by the Ministry of Earth Sciences (Newton-Bhaba Project by MoES [India] - NERC[UK]), Government of India vide grant no. MoES/NERC/IA-SWR/P2/09/2016-PC-II(i). The authors are also grateful to Mathew Rodell and Sean Swenson who helped in further understanding of level-3 GRACE dataset (personal email communication). The GRACE land are available at http://grace.jpl.nasa.gov, supported by the NASA MEaSUREs Program. The gridded rainfall 10 data were obtained from the India Meteorological Department and the well depth data from Central Ground Water Board. Details of these sources are provided in references along with their weblinks.

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

**Table-1** Food grain production and groundwater development statistics of the sub-regions.

| States | *Production (in million tonnes) | *Percentage Share of all India Production | #Population (in millions) | #Density (persons/ km²) | **Stage of Groundwater Development (in ) |
|---|---|---|---|---|---|
| Uttar Pradesh | 50.05 | 18.90 | 199.81 | 829 | 72 |
| Punjab | 28.90 | 10.92 | 27.74 | 895 | 170 |
| Rajasthan | 18.30 | 6.91 | 68.54 | 200 | 135 |
| Haryana | 16.97 | 6.41 | 25.35 | 879 | 127 |
| Maharashtra | 13.92 | 5.26 | 112.37 | 365 | 50 |
| Bihar | 13.15 | 4.97 | 104.09 | 1106 | 43 |

Source: *Directorate of Economics and Statistics, Department of Agriculture and Cooperation, 2013-14; #Census 2011; Groundwater Year Book 2013-14, CGWB.

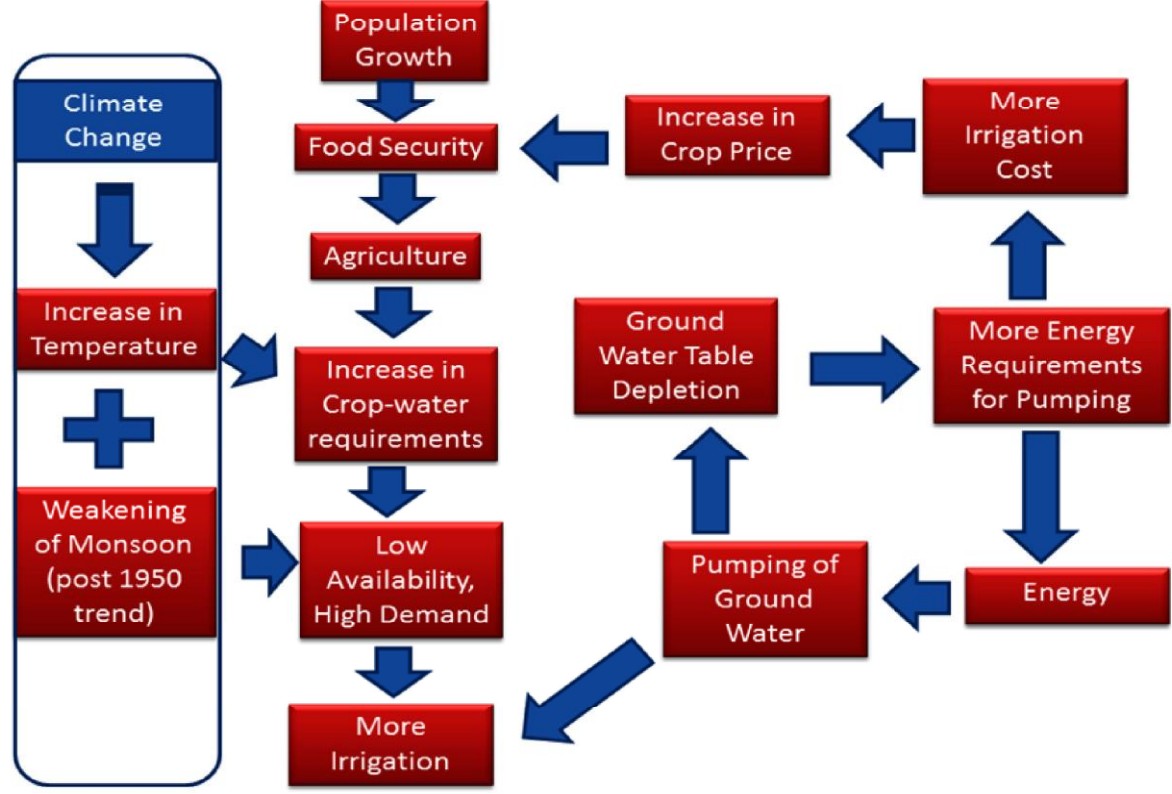

**Fig. 1.** Water-Food-Energy Nexus in India in a Changing Climate

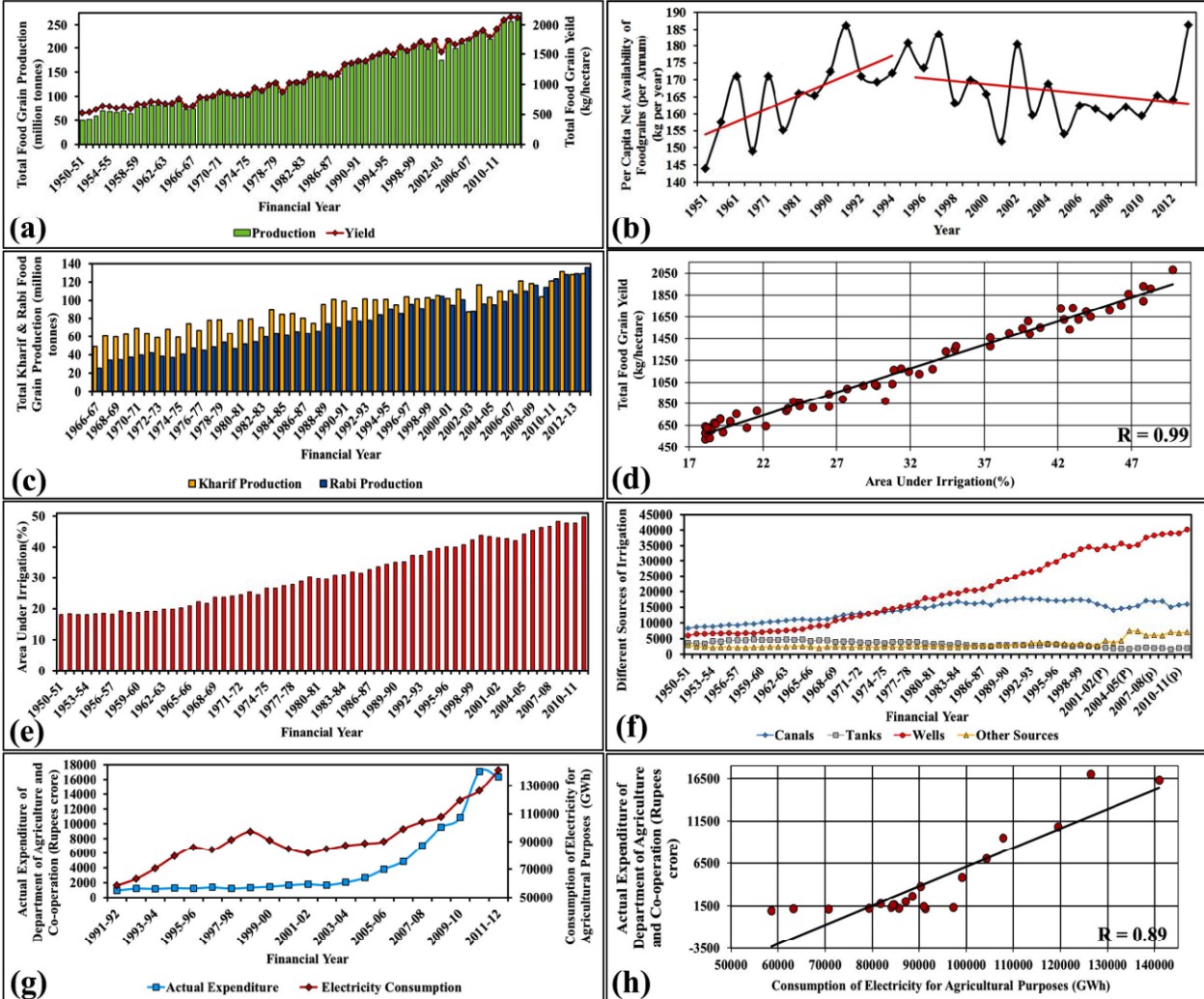

**Fig. 2.** Overview of Indian agricultural scenario. **(a)** Total food production and yield. **(b)** Per capita net availability of food grains (per annum). **(c)** Total Kharif and Rabi food grain production. **(d)** Correlation between area under irrigation and total food grain yield. **(e)** Percentage area under irrigation. **(f)** Different sources of irrigation. **(g)** Consumption of electricity for agricultural purpose and actual expenditure of department of agriculture and co-operation. **(h)** Correlation between consumption of electricity for agricultural purpose and actual expenditure of department of agriculture and co-operation.

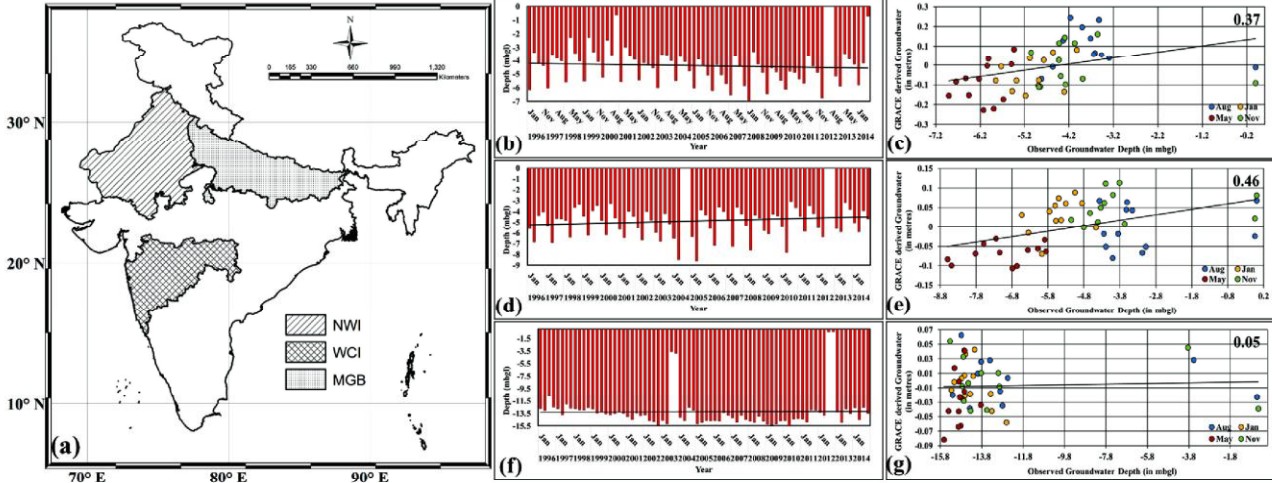

**Fig. 3.** Validation of GRACE with well observation**. (a)** Sub-regions considered for study. **(b)** Trend of groundwater levels in Middle Ganga Basin (MGB). **(c)** Correlation between observed groundwater levels and GRACE derived groundwater storage for MGB . **(d)** Trend of groundwater levels in Peninsular India. **(e)** Correlation between observed groundwater levels and GRACE derived groundwater for Peninsular India. **(f)** Trend of groundwater levels in North-Western India. **(g)** Correlation between observed groundwater levels and GRACE derived groundwater storage for North-Western India.

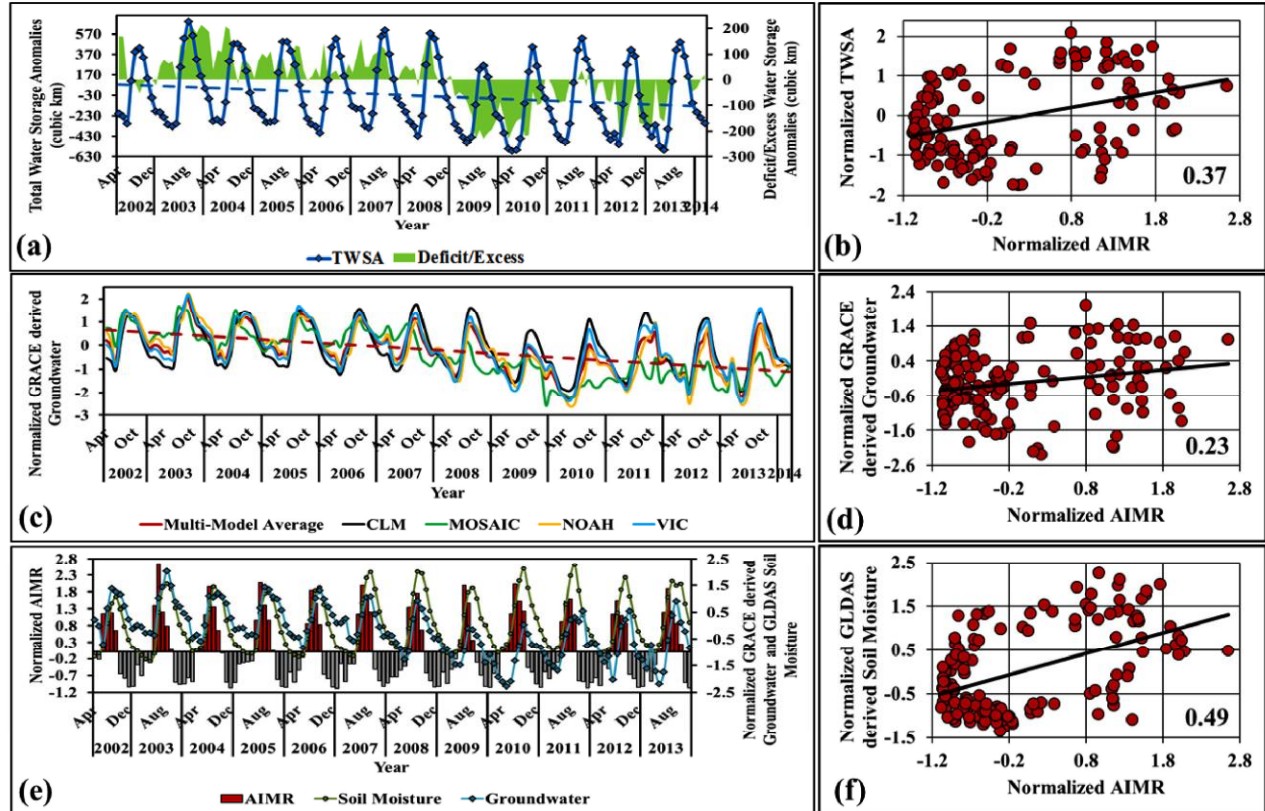

**Fig. 4.** Groundwater scenario of India. **(a)** GRACE derived total water storage anomaly. **(b)** Correlation between AIMR and TWSA. **(c)** Declining GRACE-GLDAS derived groundwater storage anomaly. **(d)** Correlation between AIMR and GRACE derived groundwater storage anomaly. **(e)** AIMR, GRACE derived groundwater anomaly and GLDAS derived soil moisture. **(f)** Correlation between AIMR and GLDAS derived soil moisture.

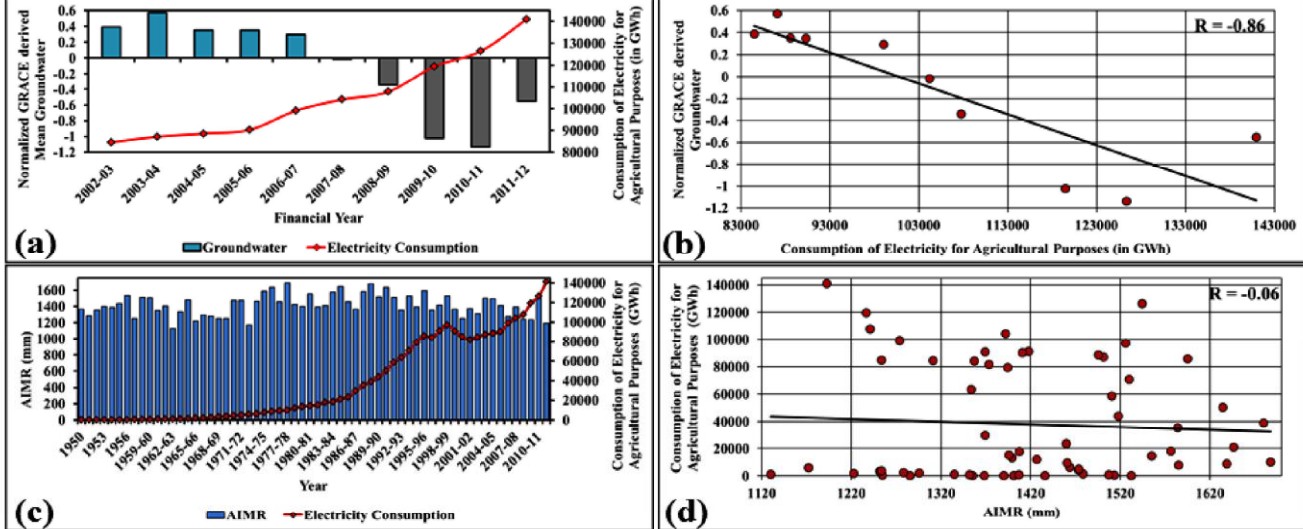

**Fig.5.** Depleting groundwater storage anomaly and increased energy consumption. **(a)** Increased electricity consumption for agricultural purpose and depletion of groundwater storage. **(b)** Scatter plot between electricity consumption for agricultural purpose and GRACE derived groundwater storage anomaly. **(c)** Time series of consumption of electricity for agricultural purposes and AIMR. **(d)** Scatter plot between AIMR and electricity consumption.

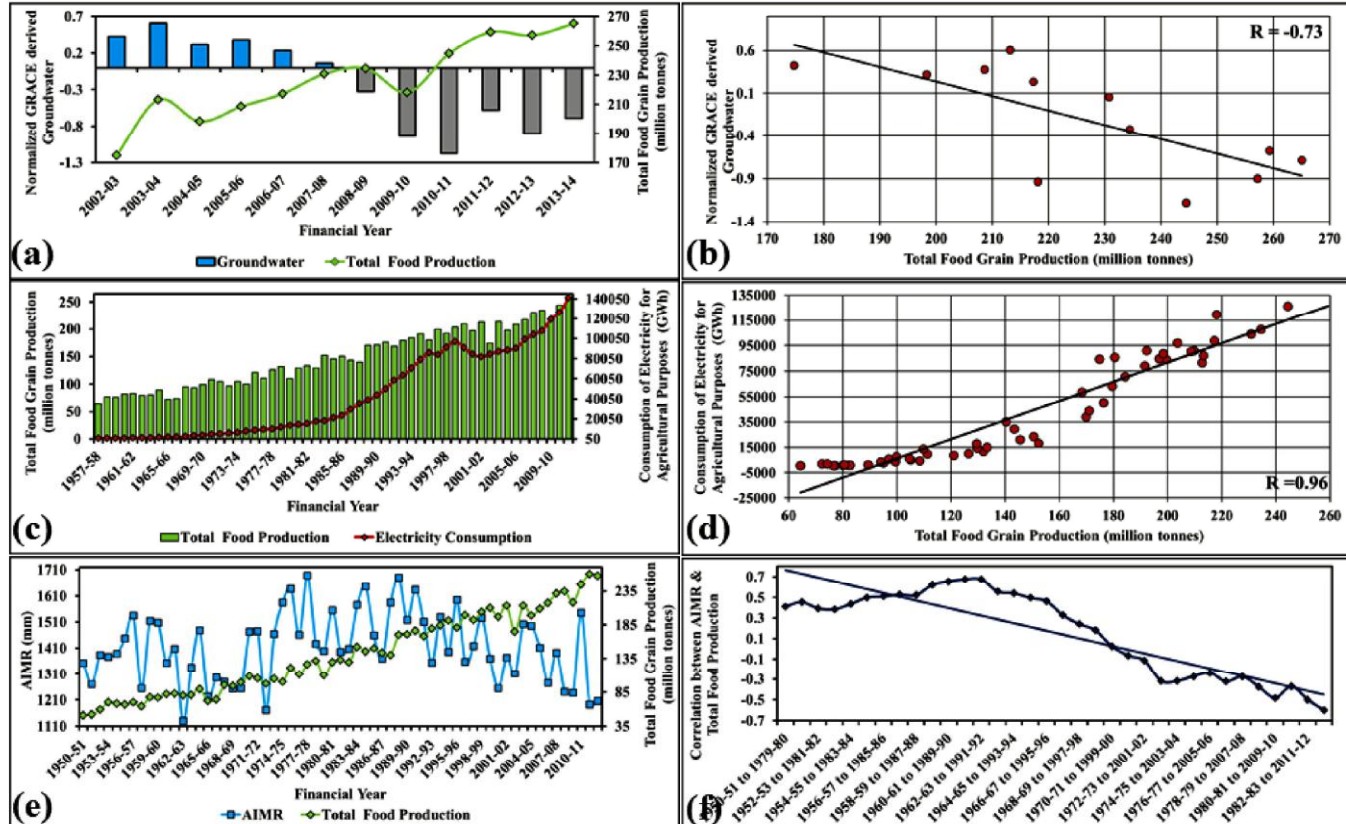

**Fig.6.** Food security maintained at the cost decreasing groundwater storage and increased electricity consumption. **(a)** Declining GRACE derived groundwater storage anomaly and increased total food production. **(b)** Scatter plot between GRACE derived groundwater storage anomaly and total food grain production. **(c)** Increased food grain production and electricity consumption for agricultural purposes. **(d)** Scatter plot between total food grain production and consumption of electricity for agricultural purposes. **(e)** Trend of AIMR and food grain production. **(f)** Trend of correlation between AIMR and total food grain production.