# Peer review of "Water Food Energy Nexus with Changing Agricultural Scenarios in India during recent Decades"

_Hydrology and Earth System Sciences, 2016_

## Short Comment (SC1) · 8 Dec 2016

After going through the abstract of the paper i find that the authors have tried well to explain the issues related to water, food and energy along with the growing demand. I propose that the term "Nexus" be replaced by some proper terminology otherwise it gives different impression, where as such relationships do exists and are obvious under the systems govern by the earth. Such study or review is of importance and throw light on the challenges being faced with growing demands.

---

## Referee Comment (RC1) · C. Scott (Referee) · 31 Jan 2017

This is a timely and relevant contribution to the scientific literature and to broader understanding of water-energy-food nexus dynamics in India.

The team has done a solid job analyzing and integrating multiple data sources. The text is, in general, well written and the figures are revealing and clearly presented.

The findings from analysis of GRACE data as well electricity usage being uncorrelated to monsoon rainfall are novel and should be highlighted.

The calls for judicious regulation are not in themselves new, but these implications of the study can be further strengthened.

With reference to page/line numbers:

1/28 - lack of correlation found between electricity usage and monsoon rainfall is crucially important and requires further explanation

1/32 - what mechanisms are proposed for judicious regulation and control?

2/29 - cite Scott, C. A., & Sharma, B.R. (2009). Energy supply and the expansion of groundwater irrigation in the Indus–Ganges Basin. International Journal of River Basin Management, 7(2), 119-124.

5/7 - while I find the selection of sub-regions to be appropriate, selection criteria should be described

7/16 - this is not my expertise area, but very good to see that validation has been conducted

9/3 - expand further on "A grave situation has been created where food security is maintained at the expense of ground water and energy security" – very important point

11/12 - this section is well done

throughout the manuscript be consistent with usage of groundwater (single, compound word), not ground water (as two separate words)

16/8 - this discussion of groundwater, cropping (cash crops vs. staples), etc. should be related to (and cited as): Shah, T., Scott, C.A., & Buechler, S.J. (2004). Water sector reforms in Mexico: Lessons for India's new water policy. Economic and Political Weekly, 361-370.

The tables and figures are well done.

The supplement is perhaps too extensive, but for online dissemination space is no constraint, so this should be retained.

---

## Referee Comment (RC2) · Anonymous Referee #2 · 2 Feb 2017

Review

A. The choice of the Topic:

(i) the authors have tried to link the food-energy security with groundwater availability based on past data over Indian region. The study area is really important and it is important bring out/highlight the link in presence of climate stress. Having said so, it is quite disappointing the manner the topic is described. Authors can't clarify the relevance of the metric used for validation of the results. For example the computation of ground water from GRACE satellite data is really not convincing as the results barely correlate with the directly measured groundwater data Fig.3(c,e,g).

(ii) topic wise, the manuscript is an intermediate between a policy report and scientific hypothesis (e.g calculation of Groundwater from GRACE vs comparing the growth of

agriculture in recent decade). I believe the scientific rigorousness of the presentation is missing (and is compromised while writing). HESS mostly is a scientific reporting journal and not a policy reporting journal, hence manuscript is not suitable for publication in this journal or else require a thorough revision. The revision may be focused on explaining a particular science topic (e.g. evaluation of GRACE derived GW data with other observation)

B. Presentation

(iii) The manuscript is very poorly written and it lacks of focus, especially in the introduction where the so called food-energy nexus is introduced. The whole introduction requires a complete restructuring in order to have a clear focus.

(iv) The term "Water-Food-Energy Nexus" is really not a suitable or scientific term to use; especially "nexus" is term quite confusing as the word implies a causative connection which is not reflected here. The study at best hypothesizes a chain of unconnected events linked together. For example, from Fig.1, the "increase in food price" is not always directly linked to "food security". Similarly "pumping of ground water" is not always linked to "irrigation". The whole chain in fig.1 is very qualitative.

C. Results and DATA

(v) estimation of ground water based on Grace and GLDASS model result is not convincing as it includes many uncertainties (which authors themselves has agreed). There is no effort made by the authors to quantify the uncertainties.

(vi) A lot of trends has been plotted. Firstly "http://eands.dacnet.nic.in/PDF/Pocket-Book2014.pdf/" is not accessible online on 2/2/2017 when I have reviewed the manuscript. Even if it is available I am not quite sure how such a data can be trusted for scientific studies. It could be good for policy "outlooks" only but unless some peer reviewed publications are available some of the results are not acceptable. For example refer Fig.6c. How do you think that there is any causal relationship between total

food production and electricity consumption? How Electricity consumption is separated from agricultural use vs. other rural use e.g. commercial agro based industries (sans requirement of irrigation) use?.

(vii) Just providing correlation and trends does not show some causative evidence (may be it is useful for social references)

(ix) Fig.4(b) and (d) is a statement of unreliability of the GRACE data in correlating with AIMR for the link you are trying to establish.

F. Conclusion

Like introduction, the section on conclusion also has no new scientific result.Even no new policy decision support system has been suggested (those which are suggested are already in place).

---

## Referee Comment (RC3) · Anonymous Referee #3 · 8 Feb 2017

**1   General comments**

The work touches on a very interesting and relevant topic. I am not familiar with hydro-logical datasets such as the ones derived from GRACE but it appears that the presentation of groundwater storage from this information is a novel and valuable contribution. The shift to irrigated crops and the link with monsoon rainfall and groundwater depletion is also interesting.

If the story regarding the nexus is simple and linear (population growth leads to increased food demand as well as more farmers, needing to irrigate more and using more electricity for that), then it is presented in too complicated a way.

The language needs to be improved and there are many technical errors which need

to be corrected.

**2 Specific comments**

- The focus on agriculture is not apparent from the title and could be included.

- The link with electricity and food production is useful and yields insights with regard to the increased role of groundwater pumping in the absence of sufficient rainfall. Some assumptions are not explicitly stated however, e.g. no estimate is given of how much of agricultural electricity use is for pumping and how that changes over time. This makes the correlations less convincing as giving strong evidence of the relationships claimed in the manuscript.

- There is a positive feedback between groundwater depletion and electricity consumption for pumping which is hinted at but not conclusively illustrated. This implies a progressive relationship between the groundwater table depth and electricity consumption for pumping. If the data allow demonstrating this, it would be a valuable addition to this research.

- The manuscript is not easy to read because of long composite sentences, poor language (omission of words, contaminations, singular/plural correspondence errors, split composite words, word placement), and unclear references. It would be correct and kind to the reader to correct all of these. I suggest having the final version proofread.

- URLs in the body should go into footnotes if allowed or in the references section.

- I suggest the use of vector graphics where possible.

- It would be good to define food security, water security and energy security.

[Figure]

- The summary and conclusions section must not contain new information but it does.

- The references are not formatted consistently, e.g. 'Pai D.S.', 'Panda, D. K.' and 'Pande S' are consecutive and all different.

- Line-by-line comments:

  – 2/16: 'water used as hydro electricity to generate power' does not make sense. Rather say e.g. 'water used for hydropower to generate electricity'
  – 4/9: 'depletion of ground water table' - 'falling groundwater table' or just 'depletion of groundwater'
  – 8/16: rate of population *growth*
  – 9/1: something is wrong with figures 2g and 2h: the 50000-130000 scale corresponds to electricity consumption in 2g and to expenditure in 2h.
  – 9/10-11: 'zero is the surface level denoting no change in the water table': this confused me - how does the surface level affect changes in the water table?
  – 15/31-32: this claim is incorrect. The study involves but does not encompass all three sectors of water, food and energy. This study refers only to fractions of the energy and water sectors.

**3  Technical corrections**

The various spelling and grammatical errors are not listed here.

- 5/7-10: inconsistent formatting of numbers, and decimals completely superfluous.

- general: 'well depth'

- 15/16: 'Validation of satellite derived groundwater' - add 'data' at the end of this

- Fig. 3a: coordinates right and bottom are duplicate and unnecessary.

- Fig. 4a: 'Deficit/Excess' first in legend but on secondary axis (on the right), this is counterintuitive.

---

## Author Comment (AC2) · 16 Feb 2017

This is a timely and relevant contribution to the scientific literature and to broader understanding of water-energy-food nexus dynamics in India.

The team has done a solid job analyzing and integrating multiple data sources.

The text is, in general, well written and the figures are revealing and clearly presented. The findings from analysis of GRACE data as well electricity usage being uncorrelated to monsoon rainfall are novel and should be highlighted. The calls for judicious regulation are not in themselves new, but these implications of the study can be further strengthened.

We sincerely thank Prof. C. Scott, reviewer 1 for reviewing our manuscript and appreciate his efforts in providing comments for the improvements of our contribution. Here, we present our responses to the comments.

1/28 - lack of correlation found between electricity usage and monsoon rainfall is crucially important and requires further explanation

We sincerely thank the reviewer for pointing this out and we agree that this is a very important finding in the context of water-food-energy nexus that needs further explanation. The new additions will be made in the revised manuscript in Sect.4.3.2 as the following:

To test the hypothesis that normal or excess monsoon years should have a lesser energy consumption (due to lesser groundwater pumping), we present a scatter plot (Fig. 5c) between AIMR and electricity consumption for agricultural purposes. Energy use in agriculture has two major usages: pumping groundwater for irrigation (electricity) (Scott and Shah, 2004; Birner et al., 2007; Kumar et al.,2013) and mechanisation (diesel) due to the use of tractors (Jha et al., 2012). Nearly 83 percent of the available water resources is used for agricultural activity, wherein 91 percent of the groundwater abstracted is used for irrigation purposes (CGWB, 2014). The agricultural electricity tariffs in India have been kept low, keeping in mind the poor economic status of farmers to facilitate groundwater pumping (Badiani et al., 2012). Due to low tariff, farmers have considered groundwater as a continuous affordable source of freshwater leading to an uncontrolled use of the same even if there is a good amount of monsoon rainfall in a specific year. The irrigation is no longer agricultural demand driven but rather dependant on the availability of electricity at a lower tariff. The scatter plot between AIMR and electricity consumption for agricultural purposes represent the same with no statistically significant correlation between them. Further to this, the irrigation practice in India is mostly flood irrigation that has very poor irrigation efficiency. This has significant implications in terms of not only agricultural water management policy but also on hydrological simulations and modelling studies. Traditionally, state of art land surface models does not consider irrigation and even if they consider irrigation, the water use is demand driven. The situation of agricultural water use in India is far from the model assumptions and hence model driven studies often underestimate the agricultural water use and groundwater abstraction. Model derived groundwater abstraction shows high dependability on monsoon rainfall (Asoka et al., 2017); however, the same needs to be tested further with ground truth. Implications of uncontrolled flood irrigation in India have been reported by Devineni et al. (2013) and Fishman et al. (2015) and this needs to be further explored to understand and realistically simulate the water cycle of Indian subcontinent.

1/32 - what mechanisms are proposed for judicious regulation and control?

We sincerely thank the reviewer and we will add the following paragraph in Section 5.

We will elaborate on the following proposed methodologies that may be adopted for judicious regulation and control.

- **Soil Moisture Monitoring and irrigation practice:** Uncontrolled use of groundwater for irrigation attributes to the lack of monitoring of soil moisture and hence the irrigation practices in India are not demand driven. Recent studies (Devineni et al., 2012; Devineni et al., 2013 and Fishman et al., 2015) show that the application of irrigation based on soil moisture condition may result into conversion of significant area from water stressed to water surplus. Further to this, the irrigation practice in India is largely flood irrigation, which has very low efficiency. Changing of irrigation type from flood to drip may reduce significant water wastage and improve irrigation efficiency.

- **Considering seasonal and extended range forecast for better water consumption:** In India, the agricultural water management models exists in theory, but they are seldom used in practice. An agricultural water allocation model at a fortnight scale or at a seasonal scale considering the improved seasonal (Saha et al., 2012) and extended range forecast (Sahai et al., 2013; Shah et al.,2016) may be useful and this needs to be further explored in near future.

- **Tariff determination based on seasonal prediction (varying volume of water usage):** Based upon seasonal prediction, a farmer would be made aware of a good/bad/normal monsoon year. Demands should be calculated based on seasonal prediction and accordingly the required amount of groundwater irrigation may be obtained considering possible margin of errors from hind-cast data. This amount may be considered as an upper limit for the famers to be used freely or at a subsidized tariff. If this upper limit is crossed, then they would not be allowed to enjoy free water and cheap electricity. This would majorly reduce the over-exploitation of groundwater.

- **Groundwater usage metering:** Electricity has been made available at a subsidised rate (almost free), increasing the accessibility of groundwater (Badiani et al.,2012; Fishman et al., 2015). If a volumetric tariff on groundwater (Shah et al., 2004) usage at a local level or at least at a district level is put in place it would also help in estimating the groundwater abstraction and accordingly management of the same.

- **Water allocation and water pricing:** Water allocation system needs to be put in place depending on the productivity and yield of crops (Huh and Lall, 2013). Water allocation and pricing should follow an optimal cropping policy (Devineni et al.,2012). Cultivation should be carried out in an area where it is best suited for a specific crop. Hence, reducing the pressure on the resources to produce the crop.

- **More production of food grains rather than cash crops:** Farmers have an inclination of producing cash crops than food grains as they are economically more lucrative. Food grains have a minimum price policy that is fixed by the government, hence profit maximisation is low. Moreover, most of the cash crops used are water intensive. Policy interventions are required to manage the same for a sustainable water and food security.

2/29 - cite Scott, C. A., & Sharma, B.R. (2009). Energy supply and the expansion of groundwater irrigation in the Indus˘AR˘Ganges Basin. International Journal of River Basin Management, 7(2), 119-124.

We thank the reviewer and we will include this very important reference in Section 1.

5/7 - while I find the selection of sub-regions to be appropriate, selection criteria should be described

We will now mention the selection criteria of each of the sub-region in Sect 2. Table-R1 presents the same (will be modified in the manuscript) stressing on the need to analyze these sub-regions further.

Table R1 shows the major food grain producing states and their percentage share to all India production. These states are among the major contributors to the total food grain production for the entire country from the northern and central part of India. It is important to note that these states face a crippling water-crisis (CGWB, 2014), either due to a bad monsoon or due to the over-use and misuse of water. They are also among the most densely populated states of India (Census 2011). Hence, the problems not only pertain to climate variability but also with the population (FAO, 2014). Thus, the following sub-regions have been selected based upon their contribution towards the total food-grain production for India, population density (increasing pressure of food security) and the stage of groundwater development (higher percentage of development implies that the consumption has exceeded recharge).

**Table R1** Food grain production and groundwater development statistics of the sub-regions.

| States | *Production (in million tonnes) | *Percentage Share of all India Production | #Population (in millions) | #Density (persons/ km²) | **Stage of Groundwater Development (in percent) |
|---|---|---|---|---|---|
| Uttar Pradesh | 50.05 | 18.90 | 199.81 | 829 | 72 |
| Punjab | 28.90 | 10.92 | 27.74 | 895 | 170 |
| Rajasthan | 18.30 | 6.91 | 68.54 | 200 | 135 |
| Haryana | 16.97 | 6.41 | 25.35 | 879 | 127 |
| Maharashtra | 13.92 | 5.26 | 112.37 | 365 | 50 |
| Bihar | 13.15 | 4.97 | 104.09 | 1106 | 43 |

Sources: *Directorate of Economics and Statistics, Department of Agriculture and Cooperation, 2013-14; #Census 2011; Groundwater Year Book 2013-14, CGWB.

7/16 - this is not my expertise area, but very good to see that validation has been conducted

We thank the reviewer for appreciating our work on validating GRACE data.

9/3 - expand further on "A grave situation has been created where food security is maintained at the expense of ground water and energy security" – very important point.

This statement has been made in relation to the explanation of Figure-1 in Sect 4.1, which we have revised as follows:

During the drought year of 2002-03, the food production was reduced by nearly 38 million tonnes as compared to the average production, but during the drought year of 2009-10, it the reduction was only by 16 million tonnes. This comes as a surprise as to what caused the restrained fall of total food grain production, despite a severe drought year. Area under irrigation has risen considerably from 18% in 1950 to nearly 50% in 2011-12. This increase in percentage of irrigated land has a high correlation with the total food grain production (R=0.99), the detrended correlation between the two is 0.46 as well. This is probably the reason behind a smaller drop in food production during the severe drought year 2009-10, as compared to other drought year 2002-03. Figure 2f shows the different sources of irrigation and the contribution of groundwater stands out clearly. CGWB (2014) report states that nearly 91 percent of the groundwater abstracted is used for irrigation alone. Sufficient groundwater for irrigation is made available to the farmers, along with energy with low or no tariff, no matter to what depth the groundwater may fall (Mukherji and Shah, 2005; Badiani et al., 2012; Fishman et al., 2015; Zaveri et al., 2016). Such schemes were in place during Green Revolution and there have been no major changes in them till date (Mukherji and Shah, 2005; Badiani et al., 2012; Fishman et al., 2015; Zaveri et al., 2016). The brunt of the availability of subsidised energy is borne by the Government of India (GoI) (Kumar, 2005; Rattan and Biswas, 2014). This causal relationship is clearly observed through the positive correlation (R=0.89) between the actual expenditure by the Department of Agriculture & Co-operation and the electricity used for agricultural purposes (Fig. 2g and h). This implies that with the increase in electricity consumption (pumping of groundwater), the expenditure incurred for agriculture also increases. Hence, a grave situation has been created where food security is maintained at the expense of groundwater and energy security (Rosegrant and Cline, 2003).

11/12 - this section is well done

We sincerely thank the reviewer for appreciating our work in Sect.4.3.2.

throughout the manuscript be consistent with usage of groundwater (single, compound word), not ground water (as two separate words)

We thank the reviewer for pointing out the inconsistency of usage of the term groundwater/ground water, it has been rectified to 'groundwater' now.

16/8 - this discussion of groundwater, cropping (cash crops vs. staples), etc. should be related to (and cited as): Shah, T., Scott, C.A., & Buechler, S.J. (2004). Water sector reforms in Mexico: Lessons for India's new water policy. Economic and Political Weekly, 361-370.

We will be incorporating the above in Section 5, where we will mention about possible judicious regulation and control.

---

## Author Comment (AC4) · 16 Feb 2017

**1 General comments**

The work touches on a very interesting and relevant topic. I am not familiar with hydrological datasets such as the ones derived from GRACE but it appears that the presentation of groundwater storage from this information is a novel and valuable contribution.

We sincerely thank the anonymous reviewer 3 for reviewing our manuscript, appreciating our analysis, and providing valuable suggestions for its further improvement.

The shift to irrigated crops and the link with monsoon rainfall and groundwater depletion is also interesting. If the story regarding the nexus is simple and linear (population growth leads to increased food demand as well as more farmers, needing to irrigate more and using more electricity for that), then it is presented in too complicated a way. The language needs to be improved and there are many technical errors which need to be corrected.

We thank the reviewer for the above mentioned comment. We agree that nexus is not so simple, as there are multiple factors that affect the system. However, here we specifically focus on irrigation driven agriculture, associated ground water pumping demanding more energy with the resulting severe depletion of groundwater. We use the satellite, on site and government data to prove our hypothesis. We tried to present the complexity of the system and hence the presentation may appear complicated. We will modify the manuscript by properly linking different paragraphs and sections to make it more easily readable. We will also correct the errors as per the reviewers' suggestions.

**2 Specific comments**

• The focus on agriculture is not apparent from the title and could be included.

We will change the title to "Water Food Energy Nexus with Changing Agricultural Scenarios in India during recent Decades". We will also add following discussion to keep the focus on agriculture.

Indian agricultural production is divided into food crops and non-food crops. The essential food crops comprise of cereals (rice, wheat, bajra, maize, millets) and pulses (tur/arhar, gram). These are considered as the staple food for nearly the entire country. The non-food crops refer to oil seeds, cotton, tobacco to name a few, these are important from the perspective of the economy to generate revenue. Hence, it is important to maintain a balance between the two, with priority given to food crops. Following table R1 shows a decadal variation of the area under cultivation, production, and percentage area under irrigation for food grains and cash crops.

**Table R1**: Decadal variation of food grains and major cash crops of India.

| Year | Food Grains | | | Major Cash Crops | | | | | | | | |
|---|---|---|---|---|---|---|---|---|---|---|---|---|
| | | | | Oilseeds | | | Cotton | | | Sugarcane | | |
| | A | P | Area Under Irrigation (%) | A | P | Area Under Irrigation (%) | A | P | Area Under Irrigation (%) | A | P | Area Under Irrigation (%) |
| 1950-51 | 97.32 | 50.82 | 18.1 | 10.73 | 5.16 | NA | 5.88 | 3.04 | 8.2 | 1.71 | 57.05 | 67.3 |
| 1960-61 | 115.58 | 82.02 | 19.1 | 13.77 | 6.98 | 3.3 | 7.61 | 5.6 | 12.7 | 2.42 | 110 | 69.3 |
| 1970-71 | 124.32 | 108.42 | 24.1 | 16.64 | 9.63 | 7.4 | 7.61 | 4.76 | 17.3 | 2.62 | 126.37 | 72.4 |
| 1980-81 | 126.67 | 129.59 | 29.6 | 17.6 | 9.37 | 14.5 | 7.82 | 7.01 | 27.3 | 2.67 | 154.25 | 81.2 |
| 1990-91 | 127.84 | 176.39 | 35.1 | 24.15 | 18.61 | 22.9 | 7.44 | 9.84 | 32.9 | 3.69 | 241.05 | 86.9 |
| 2000-01 | 121.05 | 196.81 | 43.3 | 22.77 | 18.44 | 23 | 8.53 | 9.52 | 34.3 | 4.32 | 295.96 | 92.1 |
| 2010-11 | 126.67 | 244.49 | 47.8 | 27.22 | 32.48 | 25.1 | 11.24 | 33 | 33.8 | 4.88 | 342.38 | 92.5 |

Source: Directorate of Economics and Statistics, Department of Agriculture and Cooperation, 2013-14

A=Area under cultivation (in million hectares)

P = Production (in million tonnes)

[Figure]

**Fig.R1.** Varying production (top) and percentage of area under irrigation (bottom) for food grain and cash crops.

The above Fig R1 brings out the disparity in production and percentage area of irrigation for food grains and cash crops (specifically, sugarcane) very clearly. However, the area under cultivation is largest for food grains, despite that sugarcane has higher productivity.

• The link with electricity and food production is useful and yields insights with regard to the increased role of groundwater pumping in the absence of sufficient rainfall. Some assumptions are not explicitly stated however, e.g. no estimate is given of how much of agricultural electricity use is for pumping and how that changes over time. This makes the correlations less convincing as giving strong evidence of the relationships claimed in the manuscript.

We sincerely thank the reviewer for this very important point and we agree that this has not really come out well in the previous version of the manuscript. We also agree that we have not mentioned about how much of agricultural electricity is used for groundwater pumping. Hence, we shall revise section 4.3.3 as the following:

A similar analysis has been conducted for food production and groundwater storage change. As the food production has increased there has been a decline in the storage of groundwater (Fig. 6a) and they have a high negative correlation (R= -0.73) as seen in the scatter plot (Fig.6b). On the other hand, a positive correlation (R = 0.96) exists between the food production and electricity consumption (Fig. 6d). This implies the high dependency of food production and ground water irrigation.

Energy usage in agriculture has two major usages: pumping groundwater for irrigation (electricity) (Scott and Shah, 2004; Birner et al., 2007; Kumar et al.,2013) and mechanisation (diesel) due to the use of tractors (Jha et al., 2012). Several studies have stated that the electricity for agricultural purposes is mainly used for irrigation (Scott and Shah, 2004; Birner et al., 2007; Kumar et al.,2013) because the farm mechanisation is dependent upon diesel (Jha et al., 2012). Hence, an assumption that has been considered for this analysis is that, the data for 'consumption of electricity for agricultural purposes' represent the electricity used for pumping groundwater. This will be added in the manuscript.

In Figure 6c we observe that in the year 1957-58 the electricity consumption was 544.64 GWh along with a food production of 64.31 million tonnes. 20 years later the electricity consumption increased by nearly 20 times (10107.36 GWh) but the food production increased only by twice the previous (126.41 million tonnes). In year 1997-98 electricity consumption was 97195 GWh but still the food production lagged (203 million tonnes). In the recent year of 2011-12 food production increased to 259.32 million tonnes and electricity consumption was 140960 million tonnes. Hence, over the past 54 years (1957-58 to 2011-12) electricity consumption has increased by more than 250 folds whereas the food production has increased only by four folds. Thus, when actual observed values are considered we see that electricity consumption has increased in leaps and bounds but food production has failed to do so, which brings out the concern regarding food security. Thus, there exists a clear dis-balance between the three facets of the nexus. This indicates that the food security has been maintained at the cost of water and energy security.

• There is a positive feedback between groundwater depletion and electricity consumption for pumping which is hinted at but not conclusively illustrated. This implies a progressive relationship between the groundwater table depth and electricity consumption for pumping. If the data allow demonstrating this, it would be a valuable addition to this research.

We thank the reviewer for stating this important finding and we will include the following analysis in the revised manuscript.

We hypothesize the progressive relationship between the groundwater table depth and electricity consumption for pumping, i.e., there exist a correlation between the annual change in ground water level (year 2-year1) and the electricity used for pumping in year2. We plot the same for the three regions, Middle Ganga Basin (MGB, Fig. R2(a)), North-West India (NWI, Fig. R2(b)) and West Central India (WCI, Fig. R21(c)). The duration of the analysis is from 1999 to 2011.

We find statistically significant positive correlation between ground water level drop and electricity consumption for MGB. However, statistically significant correlation does not exist for NWI and WCI. For WCI, this is expected and it is consistent with overall increase in ground water level that possibly attributes to judicious use of groundwater. However, a careful investigation for NWI reveals that the correlation value is dominated by two outliers (marked in red in Fig.R2(b)) of changes in ground water table depth. After removing the outliers, we obtain a very high statistically significant correlation as presented in Fig. R3.

We would also like to mention here, that the well data available from CGWB are not continuous (available for only months of Jan, May, August and November) and the sample size is also low. Under such situation, with spatially and temporally discontinuous ground observations, a high correlation may not be expected.

[Figure]

**Fig. R2.** Scatter plots showing the correlation between groundwater table depth and agricultural electricty consumption for the three sub region (a) MGB, (b)NWI and (c)WCI.

[Figure]

**Fig. R3.** Scatter plot of NWI after removing the outliers.

• The manuscript is not easy to read because of long composite sentences, poor language (omission of words, contaminations, singular/plural correspondence errors, split composite words, word placement), and unclear references. It would be correct and kind to the reader to correct all of these. I suggest having the final version proofread. We thank the reviewer for mentioning the above discrepancy in the manuscript, we will revise the entire manuscript with reframed short sentences and better language. References will be added accordingly and the final version would be proof-read.

• URLs in the body should go into footnotes if allowed or in the references section. We shall take care of this, reformat the references including the URLs.

• I suggest the use of vector graphics where possible. All the figures will be plotted as vector graphics (.eps) in the revised manuscript.

• It would be good to define food security, water security and energy security.

The following definitions would be added in Sect.1 for a better understanding of their interlink

The United Nations has defined food security and water security as the following:

 "Food Security is the condition in which all people, at all times, have physical, social and economic access to sufficient safe and nutritious food that meets their dietary needs and food preferences for an active and healthy life."

"Water security is defined as the capacity of a population to safeguard sustainable access to adequate quantities of acceptable quality water for sustaining livelihoods, human well-being, and socio-economic development, for ensuring protection against water-borne pollution and water-related disasters, and for preserving ecosystems in a climate of peace and political stability."

The International Energy Agency (IEA) defines energy security as "the uninterrupted availability of energy sources at an affordable price".

• The summary and conclusions section must not contain new information but it does.

We will remove the new information from conclusions and add to results and discussions.

• The references are not formatted consistently, e.g. 'Pai D.S.', 'Panda, D . K.' and 'Pande S' are consecutive and all different.

These inconsistencies will be rectified as the following and all other references will be cross-checked.

Pai, D.S., Sridhar, L., Rajeevan, M., Sreejith, O.P., Satbhai, N.S. and Mukhopadhyay, B.: Development of a new high spatial resolution (0.25° X 0.25°) long period (1901-2010) daily gridded rainfall data set over India and its comparison with existing data sets over the region, Mausam, 65(1), 1-18, 2014.

Panda, D. K. and Wahr, J.: Spatiotemporal evolution of water storage changes in India from the updated GRACE-derived gravity records. Water Resour. Res., 52(1), 135-149, doi:10.1002/2015WR017797, 2016.

Pande, S. and Savenije, H.H.: A sociohydrological model for smallholder farmers in Maharashtra, India, Water Resour. Res., 52(3), 1923-1947, doi:10.1002/2015WR017841, 2016.

• Line-by-line comments:
– 2/16: 'water used as hydro electricity to generate power' does not make sense. Rather say e.g. 'water used for hydropower to generate electricity'
This will be rectified too:
"Water is required for agricultural produce, energy is required to pump the water from various sources, and again water is used for hydropower to generate electricity."

– 4/9: 'depletion of ground water table' - 'falling groundwater table' or just 'depletion of groundwater'
This will be changed to 'depletion of groundwater'.

This will be rectified:

"However, the net per capita availability of food shows a decline post 1996 as shown in Fig. 2b, which brings forth the clear picture of a decline in the per capita food production, but a steady increase in the rate of population growth."

– 9/1: something is wrong with figures 2g and 2h: the 50000-130000 scale corresponds to electricity consumption in 2g and to expenditure in 2h.

This will be rectified, we apologize for the inadvertent error in the axis values.

– 9/10-11: 'zero is the surface level denoting no change in the water table':this confused me - how does the surface level affect changes in the water table?

Here we meant that the groundwater level is at the surface (no fall in the level). We have modified the sentence for better understanding as:

"Figure 3b, d and f show the groundwater depths measured in meters below ground level (mbgl), where zero refers to the groundwater at the surface (opening of the observation well) denoting no change in the water table. A time series of the four months have been plotted from 1996 to 2014."

– 15/31-32: this claim is incorrect. The study involves but does not encompass all three sectors of water, food and energy. This study refers only to fractions of the energy and water sectors.

We agree with the reviewer but our claim is more concentrated with respect to the agricultural ground water use and agricultural energy use. We have modified the sentence accordingly as:

"This present study is the first of its kind for India, encompassing all the three major sectors of water, food and energy from the perspective of the agricultural sector."

**3 Technical corrections**

We thank reviewer 3 for taking time out to point out these intricate errors. Spelling and grammatical errors will be corrected to the best of our knowledge.

The various spelling and grammatical errors are not listed here.

The listed-out errors will be addressed in the revised manuscript as the following:

• 5/7-10: inconsistent formatting of numbers, and decimals completely superfluous.

This will be rectified in the revised manuscript as:

"The three sub-regions studied here are, North-West India (NWI) (the states of Rajasthan, Punjab, Haryana and Delhi) covering an area of 437,739.14 km$^2$; Middle-Ganga Basin (MGB) (the states of Uttar Pradesh and Bihar) covering an area of 339,488.09 km$^2$ and West-Central India (WCI) (the states of Maharashtra and Goa) encompassing an area of 311,249.34 km$^2$."

This will be rectified in the revised manuscript as suggested.

• 15/16: 'Validation of satellite derived groundwater' - add 'data' at the end of this

This will be rectified in the revised manuscript.

• Fig. 3a: coordinates right and bottom are duplicate and unnecessary.

The duplicate coordinates will be removed.

• Fig. 4a: 'Deficit/Excess' first in legend but on secondary axis (on the right), this is counterintuitive.

This will be rectified by making deficit/excess as the second entry in the legend in Fig. 4a, also the same correction will be made in the supplementary material for Supplementary Fig. S2a, Fig. S5a and Fig. S8a.

References:

Agricultural statistics at a glance 2014, Ministry of Agriculture Department of Agriculture & Cooperation, Directorate of Economics & Statistics, Government of India, 2015 (http://eands.dacnet.nic.in/PDF/Agricultural-Statistics-At-Glance2014.pdf).

Birner, R., Gupta, S., Sharma, N., and Palaniswamy, N.: The political economy of agricultural policy reform in India: The case of fertilizer supply and electricity supply for groundwater irrigation, IFPRI, New Delhi, India, 2007.

Jha, G.K., Pal, S. and Singh, A.: Energy requirement for Indian Agriculture, ICAR, New Delhi, 2012.

Kumar, M.D., Scott, C.A. and Singh, O.P.: Can India raise agricultural productivity while reducing groundwater and energy use?, Int. J. of Water Resour. D.,29(4), 557-573, doi: 10.1080/07900627.2012.743957, 2013.

Scott, C. A., and Shah, T.: Groundwater overdraft reduction through agricultural energy policy: insights from India and Mexico, Int. J. of Water Resour. D., 20(2), 149-164, doi:10.1080/0790062042000206156, 2004.

---

## Author Response (AR1)

**Water Food Energy Nexus with Changing Agricultural Scenarios in India during recent Decades**

Beas Barik[1], Subimal Ghosh[1, 2, *], A Saheer Sahana[1], Amey Pathak[1], Muddu Sekhar[3]

[1]Department of Civil Engineering, Indian Institute of Technology Bombay, Mumbai – 400 076, India

[2]Interdisciplinary Program in Climate Studies, Indian Institute of Technology Bombay, Mumbai – 400 076, India

[3]Department of Civil Engineering, Indian Institute of Science, Bangalore – 560 012, India

*Correspondence to: Subimal Ghosh (subimal@civil.iitb.ac.in)

Response to the Short Comment by Dr. Devesh Walia

After going through the abstract of the paper i find that the authors have tried well to explain the issues related to water, food and energy along with the growing demand. I propose that the term "Nexus" be replaced by some proper terminology otherwise it gives different impression, where as such relationships do exists and are obvious under the systems govern by the earth. Such study or review is of importance and throw light on the challenges being faced with growing demands.

We sincerely thank Prof. D Walia for appreciating our work and his comment. The word "Nexus" refers to causative connection and here we analyse the same for the changing patterns of water, food and energy scenarios with linkages between them. Hence, we keep the word "nexus" in the title.

Responses to Prof. C. Scott Reviewer 1:

This is a timely and relevant contribution to the scientific literature and to broader understanding of water-energy-food nexus dynamics in India. The team has done a solid job analyzing and integrating multiple data sources. The text is, in general, well written and the figures are revealing and clearly presented. The findings from analysis of GRACE data as well electricity usage being uncorrelated to monsoon rainfall are novel and should be highlighted. The calls for judicious regulation are not in themselves new, but these implications of the study can be further strengthened.

We sincerely thank Prof. C. Scott, reviewer 1 for reviewing our manuscript and appreciate his efforts in providing comments for the improvements of our contribution. Here, we present our responses to the comments.

1/28 - lack of correlation found between electricity usage and monsoon rainfall is crucially important and requires further explanation.

We sincerely thank the reviewer for pointing this out and we agree that this is a very important finding in the context of water-food-energy nexus that needs further explanation. The new additions have been made in the revised manuscript in Sect.4.3.2 (pg12/ln24-39 and pg13/ln1-6) as the following:

To test the hypothesis that normal or excess monsoon years should have a lesser energy consumption (due to lesser groundwater pumping), we present a scatter plot (Fig. 5d) between AIMR and electricity consumption for agricultural purposes. Energy use in agriculture has two major usages: pumping groundwater for irrigation (electricity) (Scott and Shah, 2004; Birner et al., 2007; Kumar et al., 2013) and mechanisation (diesel) due to the use of tractors (Jha et al., 2012). Nearly 83 percent of the available water resources is used for agricultural activity, wherein 91 percent of the groundwater abstracted is used for irrigation purposes (CGWB, 2014). The agricultural electricity tariffs in India have been kept low, keeping in mind the poor economic status of farmers to facilitate groundwater pumping (Badiani et al., 2012). Due to low tariff, farmers have considered groundwater as a continuous affordable source of freshwater leading to an uncontrolled use of the same even if there is a good amount of monsoon rainfall in a specific year. The irrigation is no longer agricultural demand driven but rather dependent on the availability of electricity at a lower tariff. The scatter plot between AIMR and electricity consumption for agricultural purposes represent the same with no statistically significant correlation between them. Further to this, the irrigation practice in India is mostly flood irrigation that has very poor irrigation efficiency. This has significant implications in terms of not only agricultural water management policy but also on hydrological simulations and modelling studies. Traditionally, state of art land surface models does not consider irrigation and even if they consider irrigation, the water use is demand driven. The situation of agricultural water use in India is far from the model assumptions and hence model driven studies often underestimate the agricultural water use and groundwater abstraction. Model derived groundwater abstraction shows high dependability on monsoon rainfall (Asoka et al., 2017); however, the same needs to be tested further with ground truth. Implications of uncontrolled flood irrigation in India have been reported by Devineni et al. (2013) and Fishman et al. (2015) and this needs to be further explored to understand and realistically simulate the water cycle of Indian subcontinent.

1/32 - what mechanisms are proposed for judicious regulation and control?

We sincerely thank the reviewer and we have added the following paragraph in Section 5 (pg18/ln1-34).

Following are few measures which could be incorporated for judicious regulation and control:

- **Soil Moisture Monitoring and irrigation practice**: Uncontrolled use of groundwater for irrigation attributes to the lack of monitoring of soil moisture and hence the irrigation practices in India are not demand driven. Recent studies (Devineni et al., 2012; Devineni et al., 2013 and Fishman et al., 2015) show that the application of irrigation based on soil moisture condition may result into conversion of significant area from water stressed to water surplus. Further to this, the irrigation practice in India is largely flood irrigation, which has very low efficiency. Changing of irrigation type from flood to drip may reduce significant water wastage and improve irrigation efficiency.

- **Considering seasonal and extended range forecast for better water consumption**: In India, the agricultural water management models exists in theory, but they are seldom used in practice. An

agricultural water allocation model at a fortnight scale or at a seasonal scale considering the improved seasonal (Saha et al., 2012) and extended range forecast (Sahai et al., 2013; Shah et al.,2016) may be useful and this needs to be further explored in near future.

- **Tariff determination based on seasonal prediction (varying volume of water usage)**: Based upon seasonal prediction, a farmer would be made aware of a good/bad/normal monsoon year. Demands should be calculated based on seasonal prediction and accordingly the required amount of groundwater irrigation may be obtained considering possible margin of errors from hind-cast data. This amount may be considered as an upper limit for the famers to be used freely or at a subsidized tariff. If this upper limit is crossed, then they would not be allowed to enjoy free water and cheap electricity. This would majorly reduce the over-exploitation of groundwater.

- **Groundwater usage metering**: Electricity has been made available at a subsidised rate (almost free), increasing the accessibility of groundwater (Badiani et al.,2012; Fishman et al., 2015). If a volumetric tariff on groundwater (Shah et al., 2004) usage at a local level or at least at a district level is put in place it would also help in estimating the groundwater abstraction and accordingly management of the same.

- **Water allocation and water pricing**: Water allocation system needs to be put in place depending on the productivity and yield of crops (Huh and Lall, 2013). Water allocation and pricing should follow an optimal cropping policy (Devineni et al.,2012). Cultivation should be carried out in an area where it is best suited for a specific crop. Hence, reducing the pressure on the resources to produce the crop.

- **More production of food grains rather than cash crops**: Farmers have an inclination of producing cash crops than food grains as they are economically more lucrative. Food grains have a minimum price policy that is fixed by the government, hence profit maximisation is low. Moreover, most of the cash crops used are water intensive. Policy interventions are required to manage the same for a sustainable water and food security.

2/29 - cite Scott, C. A., & Sharma, B.R. (2009). Energy supply and the expansion of groundwater irrigation in the Indusâ˘ARˇGanges Basin. International Journal of River Basin Management, 7(2), 119-124.

We thank the reviewer and we have included this very important reference in Section 1 (pg2/ln35)

5/7 - while I find the selection of sub-regions to be appropriate, selection criteria should be described

We have mentioned the selection criteria of each of the sub-region in Sect 2 (pg5/ln15-23). Table-R1 presents the same (will be modified in the manuscript as Table-1) stressing on the need to analyze these sub-regions further.

Table R1 shows the major food grain producing states and their percentage share to all India production. Thus, the following sub-regions have been selected based upon their contribution towards the total food-grain production for India, population density (increasing pressure of food security) and the stage of groundwater development (higher percentage of development implies that the consumption has exceeded recharge).

**Table R1** Food grain production and groundwater development statistics of the sub-regions.

| States | *Production (in million tonnes) | *Percentage Share of all | #Population (in millions) | #Density (persons/ km$^2$) | **Stage of Groundwater |
|---|---|---|---|---|---|

| | | India Production | | | Development (in percent) |
|---|---|---|---|---|---|
| Uttar Pradesh | 50.05 | 18.90 | 199.81 | 829 | 72 |
| Punjab | 28.90 | 10.92 | 27.74 | 895 | 170 |
| Rajasthan | 18.30 | 6.91 | 68.54 | 200 | 135 |
| Haryana | 16.97 | 6.41 | 25.35 | 879 | 127 |
| Maharashtra | 13.92 | 5.26 | 112.37 | 365 | 50 |
| Bihar | 13.15 | 4.97 | 104.09 | 1106 | 43 |

Sources: [*]Directorate of Economics and Statistics, Department of Agriculture and Cooperation, 2013-14; [#]Census 2011; Groundwater Year Book 2013-14, CGWB.

7/16 - this is not my expertise area, but very good to see that validation has been conducted

We thank the reviewer for appreciating our work on validating GRACE data.

9/3 - expand further on "A grave situation has been created where food security is maintained at the expense of ground water and energy security" – very important point.

This statement has been made in relation to the explanation of Figure-2 in Sect 4.1(pg9/ln3-21), which we have revised as follows:

During the drought year of 2002-03, the food production was reduced by nearly 38 million tonnes as compared to the average production, but during the drought year of 2009-10, the reduction was only by 16 million tonnes. This comes as a surprise as to what caused the restrained fall of total food grain production, despite a severe drought year. Percentage area under irrigation has risen considerably from 18 percent in 1950 to nearly 50 percent in 2011-12. This increase in percentage of irrigated land has a high correlation with the total food grain production (R=0.99), the detrended correlation between the two is 0.46 as well. This is probably the reason behind a smaller drop in food production during the severe drought year 2009-10, as compared to other drought year 2002-03. Figure 2f shows the different sources of irrigation and the contribution of groundwater stands out clearly. CGWB (2014) report states that nearly 91 percent of the groundwater abstracted is used for irrigation alone. Sufficient groundwater for irrigation is made available to the farmers, along with energy with low or no tariff, no matter to what depth the groundwater may fall (Mukherji and Shah, 2005; Badiani et al., 2012; Fishman et al., 2015; Zaveri et al., 2016). Such schemes were in place during Green Revolution and there have been no major changes in them till date (Mukherji and Shah, 2005; Badiani et al., 2012; Fishman et al., 2015; Zaveri et al., 2016). The brunt of the availability of subsidised energy is borne by the Government of India (GoI) (Kumar, 2005; Rattan and Biswas, 2014). This causal relation is clearly observed through the positive correlation (R=0.89) between the actual expenditure by the department of agriculture and co-operation and the electricity used for agricultural purposes (Fig. 2g and h). This implies that with the increase in electricity consumption (pumping of groundwater), the expenditure incurred for agriculture also increases. Hence, a grave situation has been created where food security is maintained at the expense of groundwater and energy security (Rosegrant and Cline, 2003).

11/12 - this section is well done

We sincerely thank the reviewer for appreciating our work in Sect.4.3.2.

throughout the manuscript be consistent with usage of groundwater (single, compound word), not ground water (as two separate words)

5   We thank the reviewer for pointing out the inconsistency of usage of the term groundwater/ground water, it has been rectified to 'groundwater' in the revised manuscript.

16/8 - this discussion of groundwater, cropping (cash crops vs. staples), etc. should be related to (and cited as): Shah, T., Scott, C.A., & Buechler, S.J. (2004). Water sector reforms in Mexico: Lessons for India's new water policy. Economic and Political Weekly, 361-370.

10   We have incorporated the above in Section 5 (pg18/ln24), where we have mentioned about possible judicious regulation and control.

Responses to Anonymous Referee 2:

We sincerely thank the anonymous reviewer 2 and appreciate his/ her efforts in providing critical comments for the improvements of our contribution. Here, we present our responses to the comments.

15   A. The choice of the Topic:

(i) the authors have tried to link the food-energy security with groundwater availability based on past data over Indian region. The study area is really important and it is important bring out/highlight the link in presence of climate stress. Having said so, it is quite disappointing the manner the topic is described. Authors can't clarify the relevance of the metric used for validation of the results. For example, the computation of ground water from
20   GRACE satellite data is really not convincing as the results barely correlate with the directly measured groundwater data Fig.3(c, e, g).

We thank the reviewer for pointing out the importance of the case study area.

We appreciate the comments related to the validation of GRACE derived groundwater storage information. Firstly, we would like to present a series of scientific studies that have used the GRACE derived water storage
25   data for the case study region. This table has been added as a part of the Supplementary Material (as Table S2).

**Table R2** Studies conducted using GRACE over the Indian region

| Reference | Case-study Region | Source |
|---|---|---|
| Rodell, M., Velicogna, I. and Famiglietti, J. S.: Satellite-based estimates of groundwater depletion in India., *Nature*, 460(7258), 999–1002, doi:10.1038/nature08238, 2009. | North-West India (Punjab, Haryana, Delhi, Rajasthan) | http://www.nature.com/nature/journal/v460/n7258/pdf/nature08238.pdf |

| | | |
|---|---|---|
| Tiwari, V. M., Wahr, J. and Swenson, S.: Dwindling groundwater resources in northern India, from satellite gravity observations, *Geophys. Res. Lett.*, 36(18), 1–5, doi:10.1029/2009GL039401, 2009. | Indus-Ganga-Brahmaputra Basin. | http://onlinelibrary.wiley.com/doi/10.1029/2009GL039401/epdf |
| Chen, J., Li, J., Zhang, Z. and Ni, S.: Long-term groundwater variations in Northwest India from satellite gravity measurements, *Glob. Planet. Change*, 116, 130–138, doi:10.1016/j.gloplacha.2014.02.007, 2014. | North-West India | http://ac.els-cdn.com/S0921818114000526/1-s2.0-S0921818114000526-main.pdf?_tid=b0e17334-e9d5-11e6-9328-00000aab0f02&acdnat=1486101648_83a007a750a5b266c5a402b016f1bfa2 |
| Dasgupta, S., Das, I. C., Subramanian, S. K. and Dadhwal, V. K.: Space-based gravity data analysis for groundwater storage estimation in the Gangetic plain, India, *Curr. Sci.*, 107(5), 832–844, 2014. | Gangetic Plain (Uttar Pradesh, Bihar, West Bengal) | http://www.currentscience.ac.in/Volumes/107/05/0832.pdf |
| Prakash, S., Gairola, R. M., Papa, F. and Mitra, A. K.: An assessment of terrestrial water storage, rainfall and river discharge over Northern India from satellite data, *Curr. Sci.*, 107(9), 1582–1586, 2014. | India | http://www.currentscience.ac.in/Volumes/107/09/1582.pdf |
| Khandu, Forootan, E., Schumacher, M., Awange, J. L. and Müller Schmied, H.: Exploring the influence of precipitation extremes and human water use on total water storage (TWS) changes in the Ganges-Brahmaputra-Meghna River Basin, *Water Resour. Res.*, doi:10.1002/2015WR018113, 2016. | Ganga-Brahmaputra-Meghna Basin | http://onlinelibrary.wiley.com/doi/10.1002/2015WR018113/epdf |
| Yi, S., Sun, W., Feng, W. and Chen, J.: Anthropogenic and climate-driven water depletion in Asia, *Geophys. Res. Lett.*, doi:10.1002/2016GL069985, 2016. | Asia | http://onlinelibrary.wiley.com/doi/10.1002/2016GL069985/epdf |

| Panda, D. K. and Wahr, J.: Spatiotemporal evolution of water storage changes in India from the updated GRACE-derived gravity records. *Water Resources Research*, *52*(1), 135-149, 2016. | India | http://onlinelibrary.wiley.com/doi/10.1002/2015WR017797/epdf |
|---|---|---|
| Asoka, A., Gleeson T., Wada Y. and Mishra, V.: Relative contribution of monsoon precipitation and pumping to changes in groundwater storage in India. *Nature geoscience*, 10, 109–117, doi:10.1038/ngeo2869, 2017 | India | http://palgrave.nature.com/ngeo/journal/v10/n2/pdf/ngeo2869.pdf |

Further to this, we would like to respectfully point out that the well data available from CGWB are not continuous and the sample size is also low. Under such situation, with spatially and temporally discontinuous ground observations, a high correlation may not be expected. However, except one region (NWI), we have got statistically significant correlation between GRACE derived storage and well depth data (p-value < 0.05) and this justifies the use of GRACE for the present study to estimate groundwater storage.

The above discussion has been added in the revised manuscript in Sect 4.2(pg10/ln17-21).

(ii) Topic wise, the manuscript is an intermediate between a policy report and scientific hypothesis (e.g calculation of Groundwater from GRACE vs comparing the growth of agriculture in recent decade). I believe the scientific rigorousness of the presentation is missing (and is compromised while writing). HESS mostly is a scientific reporting journal and not a policy reporting journal, hence manuscript is not suitable for publication in this journal or else require a thorough revision. The revision may be focused on explaining a particular science topic (e.g. evaluation of GRACE derived GW data with other observation)

We respectfully disagree with the reviewer. We strongly feel that one of the purpose of scientific studies related to water is to help the community in planning for sustainable water management through policy reforms specifically for the regions which are severely affected by anthropogenic activities. We also would like to quote few lines from the objectives of the present special issue of HESS:

"Contributions are invited on various aspects of the study of the hydrological and hydrogeological processes, subsurface–surface–climate interactions, water resources and risks, socio-hydrological interactions, and the relationship between the water cycle and human development."

"The Ganges basin in particular exhibits extreme hydrological behaviour, including but not limited to the extent of human irrigation, the size and human use of its groundwater resources, the speed of land-use change, and the magnitude and seasonality of the Indian monsoon."

We also present an article from HESS on a similar topic but for different case study region.

Scott, Christopher A., et al. "Irrigation efficiency and water-policy implications for river basin resilience." *Hydrology and Earth System Sciences* 18.4 (2014): 1339.

Further to this, we also quote the comments from the other reviewers:

Reviewer #1:

5 "This is a timely and relevant contribution to the scientific literature and to broader understanding of water-energy-food nexus dynamics in India"

Reviewer #3:

"The work touches on a very interesting and relevant topic."

Hence, we would like to keep the focus of the manuscript unchanged.

10 B. Presentation

(iii) The manuscript is very poorly written and it lacks of focus, especially in the introduction where the so called food-energy nexus is introduced. The whole introduction requires a complete restructuring in order to have a clear focus.

In the first version of the manuscript the introduction was structured in the following way:

15 Introduction to water-food-energy nexus → Introduction to scenarios in India → Background hydro-climatic patterns in India → Literature focussing on changes in water resources → research gap and development of hypothesis.

However, we agree that there are certain limitations in the initial write up in terms of absence of definition of water food and energy security, linkages between the paragraphs etc. We have now modified the same.

20 (iv) The term "Water-Food-Energy Nexus" is really not a suitable or scientific term to use; especially "nexus" is term quite confusing as the word implies a causative connection which is not reflected here. The study at best hypothesizes a chain of unconnected events linked together. For example, from Fig.1, the "increase in food price" is not always directly linked to "food security". Similarly "pumping of ground water" is not always linked to "irrigation". The whole chain in fig.1 is very qualitative.

25 We respectfully disagree that water-food-energy nexus is not a scientific term. Here we present a list of scientific article that use the term:

**Table R3:**

| Reference | Source |
|---|---|
| Hoff H. Understanding the nexus: Background paper for the Bonn2011 Nexus Conference. | http://wef-conference.gwsp.org/fileadmin/documents_news/understanding_the_nexus.pdf |

| | |
|---|---|
| Jakob Granit , Anders Jägerskog , Andreas Lindström , Gunilla Björklund , Andrew Bullock , Rebecca Löfgren , George de Gooijer & Stuart Pettigrew (2012) Regional Options for Addressing the Water, Energy and Food Nexus in Central Asia and the Aral Sea Basin, *International Journal of Water Resources* Development, 28:3, 419-432, DOI: 10.1080/07900627.2012.684307 | http://www.tandfonline.com/doi/pdf/10.1080/07900627.2012.684307?needAccess=true |
| FAO: The Water-Energy-Food Nexus - A new approach in support of food security and sustainable agriculture, Food Agric. Organ. United Na, 1–11, doi:10.1039/C4EW90001D, 2014. | http://www.fao.org/nr/water/docs/FAO_nexus_concept.pdf |
| WADA, Christopher A; BURNETT, Kimberly; GURDAK, Jason *J. Sustainable Agriculture Irrigation Managemen*t: The Water-Energy-Food Nexus in Pajaro Valley, California. Sustainable Agriculture Research, [S.l.], v. 5, n. 3, p. p76, may 2016. ISSN 1927-0518. doi:http://dx.doi.org/10.5539/sar.v5n3p76. | http://ageconsearch.umn.edu/bitstream/241747/2/P7-p76-83.pdf |
| Endo, A., Tsurita, I., Burnett, K. and Orencio, P. M.: A review of the current state of research on the water, energy, and food nexus, *J. Hydrol. Reg. Stud.*, doi:10.1016/j.ejrh.2015.11.010, 2015. | http://ac.els-cdn.com/S2214581815001251/1-s2.0-S2214581815001251-main.pdf?_tid=38adc236-e9ef-11e6-9eec-00000aab0f01&acdnat=1486112614_472590260796ab6befe2c4830ebc7640 |
| Gurdak, J.J., Geyer, G.E., Nanus, L., Taniguchi, M., and Corona, C.R., 2016, Scale dependence of controls on groundwater vulnerability in the water-energy-food nexus, California Coastal Basin aquifer system, *Journal of Hydrology: Regional Studies*, special issue on the Water-Energy-Food Nexus of the Asia-Pacific. | http://www.sciencedirect.com/science/article/pii/S2214581816000057 |
| Hatfield-Dodds, Steve, Heinz Schandl, Philip D. Adams, Timothy M. Baynes, Thomas S. Brinsmead, Brett A. Bryan, Francis HS Chiew et al. "Australia is 'free to choose'economic growth and falling environmental pressures." *Nature* 527, no. 7576 (2015): 49-53. | http://www.nature.com/nature/journal/v527/n7576/pdf/nature16065.pdf |
| Taniguichi, M., Endo, A., Gurdak, J.J., and Swarzenski, P., 2016 - In Review, Water-Energy- | |

Food Nexus in Asia Pacific Region, *Journal of Hydrology: Regional Studies*, special issue on the Water-Energy-Food Nexus of the Asia-Pacific Region.

We agree that Figure 1 hypothesised the possible causative connection between different components of water-food-energy system specifically for Indian context and this has been further tested statistically with different datasets. Hence, the use of the word "nexus" is justified.

5     We also agree that there are multiple reasons associated with food security and one of them is increase in food price. This causal connection is relevant to water food energy nexus in India. Similarly, pumping of groundwater has multiple uses (domestic and industrial) but the major use (91percent) is in terms of irrigation (CGWB, 2014). Figure 2(f) confirms the same showing wells as the major source of irrigation for net irrigated area for all-India. The above mentioned discussions have been added in the revised manuscript in Sect.1 (pg4/ln13-14).

10     C. Results and Data

(v) Estimation of ground water based on Grace and GLDASS model result is not convincing as it includes many uncertainties (which authors themselves have agreed). There is no effort made by the authors to quantify the uncertainties.

We have agreed that significant uncertainty exist in the GRACE derived groundwater storage and have presented

15     in the earlier version in terms of bands (in Fig. 4c). We have revised the figure 4c showing individual members to explain the uncertainty along with Supplementary Figures S5c, S8c and S11c. The revised figures are presented below:

[Figure]

**Fig. R1.** Declining GRACE-GLDAS derived groundwater storage anomaly for India (modified version of Fig 4c)

[Figure]

**Fig.R2.** GRACE-GLDAS derived groundwater storage anomaly for MGB. (modified version of Supplementary Fig S5c).

[Figure]

**Fig.R3.** GRACE-GLDAS derived groundwater storage anomaly for NWI. (modified version of Supplementary Fig S8c).

[Figure]

**Fig.R4.** GRACE-GLDAS derived groundwater storage anomaly for WCI. (modified version of Supplementary Fig S11c).

The following sentences have been added for India and each of the sub-regions (MGB, NWI and WCI) in Sect. 4.3.1 (pg11/ln10-13), Sect.4.4.1 (pg14/ln21-25), Sect. 4.4.2 (pg15/ln28-30) and Sect. 4.4.3 (pg16/ln24-27) respectively to bring out the uncertainty.

Figure R1 shows the GRACE-GLDAS derived groundwater storage for India. The red line represents the multi-model average, which has a significant negative trend (p-value:$2.02\times10-13$). VIC and NOAH has the least deviation from the mean, whereas CLM and MOSAIC has a larger deviation resulting in an increased uncertainty.

Figure R2 shows the GRACE-GLDAS derived groundwater storage for MGB. Similar, to the analysis for India, here the derived groundwater also shows a statistically significant negative trend (p-value: $2.58\times10-21$). Deviations for the simulations from MOSAIC and CLM are higher than those from VIC and NOAH. Both, MOSAIC and CLM derived groundwater shows a higher deviation specifically around the year 2008-09.

Figure R3 shows the GRACE-GLDAS derived groundwater for NWI. Here as well the multi-model average line has a significant negative trend (p-value: $6.79\times10-29$). The uncertainty is low compared to the other regions.

Figure R4 shows the GRACE-GLDAS derived groundwater for WCI. The multi-model average has an increasing positive trend (p-value: $2.1\times10-7$). MOSAIC derived groundwater has a higher deviation from multi-model average through the entire study period.

(vi) A lot of trends has been plotted. Firstly "http://eands.dacnet.nic.in/PDF/PocketBook2014.pdf/" is not accessible online on 2/2/2017 when I have reviewed the manuscript. Even if it is available I am not quite sure how such a data can be trusted for scientific studies. It could be good for policy "outlooks" only but unless some peer reviewed publications are available some of the results are not acceptable. For example, refer Fig.6c. How do you think that there is any causal relationship between total food production and electricity consumption? How Electricity consumption is separated from agricultural use vs. other rural use e.g. commercial agro-based industries (sans requirement of irrigation) use?

The corrected site is http://eands.dacnet.nic.in/PDF/Pocket-Book2014.pdf

The "/" at the end of the weblink does not allow to access the website. We will correct it in the revised manuscript and sincerely apologize for this inadvertent error.

The data is from Department of Agriculture, Government of India and hence it is reliable.

We also would like to clarify that this is the agricultural electricity consumption and not total electricity consumption. This was explicitly mentioned in the earlier version of the manuscript (Abstract, line 25).

(vii) Just providing correlation and trends does not show some causative evidence (may be it is useful for social references)

Computations of correlation and trends are scientific and statistical methods to be applied for any study irrespective of the focus of the study, societal or technical.

(ix) Fig.4(b) and (d) is a statement of unreliability of the GRACE data in correlating with AIMR for the link you are trying to establish.

We would like to point that Fig. 4 (b) and (d) do not show the unreliability of the data; rather they show the importance of human interventions in the water cycle that reduce the correlation. This is a very important figure and the first reviewer has appreciated the same:

"The findings from analysis of GRACE data as well electricity usage being uncorrelated to monsoon rainfall are novel and should be highlighted."

F. Conclusion like introduction, the section on conclusion also has no new scientific result. Even no new policy decision support system has been suggested (those which are suggested are already in place)

We agree and thank the reviewer for pointing this out. We have modified the conclusions following the suggestions from all the reviewers in Section 5 (pg18/ln1-34).

Following are few measures which could be incorporated for judicious regulation and control:

- **Soil Moisture Monitoring and irrigation practice:** Uncontrolled use of groundwater for irrigation attributes to the lack of monitoring of soil moisture and hence the irrigation practices in India are not demand driven. Recent studies (Devineni et al., 2012; Devineni et al., 2013 and Fishman et al., 2015) show that the application of irrigation based on soil moisture condition may result into conversion of significant area from water stressed to water surplus. Further to this, the irrigation practice in India is largely flood irrigation, which has very low efficiency. Changing of irrigation type from flood to drip may reduce significant water wastage and improve irrigation efficiency.

- **Considering seasonal and extended range forecast for better water consumption:** In India, the agricultural water management models exists in theory, but they are seldom used in practice. An agricultural water allocation model at a fortnight scale or at a seasonal scale considering the improved seasonal (Saha et al., 2012) and extended range forecast (Sahai et al., 2013; Shah et al.,2016) may be useful and this needs to be further explored in near future.

- **Tariff determination based on seasonal prediction (varying volume of water usage):** Based upon seasonal prediction, a farmer would be made aware of a good/bad/normal monsoon year. Demands should be calculated based on seasonal prediction and accordingly the required amount of groundwater irrigation may be obtained considering possible margin of errors from hind-cast data. This amount may be considered as an upper limit for the famers to be used freely or at a subsidized tariff. If this upper limit is crossed, then they would not be allowed to enjoy free water and cheap electricity. This would majorly reduce the over-exploitation of groundwater.

- **Groundwater usage metering:** Electricity has been made available at a subsidised rate (almost free), increasing the accessibility of groundwater (Badiani et al.,2012; Fishman et al., 2015). If a volumetric tariff on groundwater (Shah et al., 2004) usage at a local level or at least at a district level is put in place it would also help in estimating the groundwater abstraction and accordingly management of the same.

- **Water allocation and water pricing:** Water allocation system needs to be put in place depending on the productivity and yield of crops (Huh and Lall, 2013). Water allocation and pricing should follow an optimal cropping policy (Devineni et al.,2012). Cultivation should be carried out in an area where it is best suited for a specific crop. Hence, reducing the pressure on the resources to produce the crop.

- **More production of food grains rather than cash crops:** Farmers have an inclination of producing cash crops than food grains as they are economically more lucrative. Food grains have a minimum price policy that is fixed by the government, hence profit maximisation is low. Moreover, most of the cash crops used are water intensive. Policy interventions are required to manage the same for a sustainable water and food security.

10 Responses to Anonymous Referee 3:

**1 General comments**

The work touches on a very interesting and relevant topic. I am not familiar with hydrological datasets such as the ones derived from GRACE but it appears that the presentation of groundwater storage from this information is a novel and valuable contribution.

15 We sincerely thank the anonymous reviewer 3 for reviewing our manuscript, appreciating our analysis, and providing valuable suggestions for its further improvement.

The shift to irrigated crops and the link with monsoon rainfall and groundwater depletion is also interesting. If the story regarding the nexus is simple and linear (population growth leads to increased food demand as well as more farmers, needing to irrigate more and using more electricity for that), then it is presented in too complicated a
20 way.
The language needs to be improved and there are many technical errors which need to be corrected.

We thank the reviewer for the above mentioned comment. We agree that nexus is not so simple, as there are multiple factors that affect the system. However, here we specifically focus on irrigation driven agriculture, associated groundwater pumping demanding more energy with the resulting severe depletion of groundwater. We
25 use the satellite, on site and government data to prove our hypothesis. We tried to present the complexity of the system and hence the presentation may appear complicated. We will modify the manuscript by properly linking different paragraphs and sections to make it more easily readable. We will also correct the errors as per the reviewers' suggestions.

**2 Specific comments**

30 • The focus on agriculture is not apparent from the title and could be included.

We have changed the title to "Water Food Energy Nexus with Changing Agricultural Scenarios in India during recent Decades". We have also added the following discussion to keep the focus on agriculture in Sect.2 (pg4/ln36-38 and pg5/ln1-3).

Indian agricultural production is divided into food crops and non-food crops. The essential food crops comprise of cereals (rice, wheat, bajra, maize, millets) and pulses (tur/arhar, gram). These are considered as the staple food for nearly the entire country. The non-food crops refer to oil seeds, cotton, tobacco to name a few, these are important from the perspective of the economy to generate revenue. Hence, it is important to maintain a balance between the two, with priority given to food crops. Following table R4 (added in Supplementary Material as Table S1) shows a decadal variation of the area under cultivation, production, and percentage area under irrigation for food grains and cash crops.

**Table R4**: Decadal variation of food grains and major cash crops of India.

| Year | Food Grains | | | Major Cash Crops | | | | | | | | |
| | | | | Oilseeds | | | Cotton | | | Sugarcane | | |
| | A | P | Area Under Irrigation (%) | A | P | Area Under Irrigation (%) | A | P | Area Under Irrigation (%) | A | P | Area Under Irrigation (%) |
|---|---|---|---|---|---|---|---|---|---|---|---|---|
| 1950-51 | 97.32 | 50.82 | 18.1 | 10.73 | 5.16 | NA | 5.88 | 3.04 | 8.2 | 1.71 | 57.05 | 67.3 |
| 1960-61 | 115.58 | 82.02 | 19.1 | 13.77 | 6.98 | 3.3 | 7.61 | 5.6 | 12.7 | 2.42 | 110 | 69.3 |
| 1970-71 | 124.32 | 108.42 | 24.1 | 16.64 | 9.63 | 7.4 | 7.61 | 4.76 | 17.3 | 2.62 | 126.37 | 72.4 |
| 1980-81 | 126.67 | 129.59 | 29.6 | 17.6 | 9.37 | 14.5 | 7.82 | 7.01 | 27.3 | 2.67 | 154.25 | 81.2 |
| 1990-91 | 127.84 | 176.39 | 35.1 | 24.15 | 18.61 | 22.9 | 7.44 | 9.84 | 32.9 | 3.69 | 241.05 | 86.9 |
| 2000-01 | 121.05 | 196.81 | 43.3 | 22.77 | 18.44 | 23 | 8.53 | 9.52 | 34.3 | 4.32 | 295.96 | 92.1 |
| 2010-11 | 126.67 | 244.49 | 47.8 | 27.22 | 32.48 | 25.1 | 11.24 | 33 | 33.8 | 4.88 | 342.38 | 92.5 |

Source: Directorate of Economics and Statistics, Department of Agriculture and Cooperation, 2013-14

A=Area under cultivation (in million hectares)

P = Production (in million tonnes)

[Figure]

**Fig.R5.** Varying production (top) and percentage of area under irrigation (bottom) for food grain and cash crops.

The above Fig R5 (added in Supplementary Material as Fig. S2) brings out the disparity in production and percentage area of irrigation for food grains and cash crops (specifically, sugarcane) very clearly. However, the area under cultivation is largest for food grains, despite that sugarcane has higher productivity.

• The link with electricity and food production is useful and yields insights with regard to the increased role of groundwater pumping in the absence of sufficient rainfall. Some assumptions are not explicitly stated however, e.g. no estimate is given of how much of agricultural electricity use is for pumping and how that changes over time. This makes the correlations less convincing as giving strong evidence of the relationships claimed in the manuscript.

We sincerely thank the reviewer for this very important point and we agree that this has not really come out well in the previous version of the manuscript. We also agree that we have not mentioned about how much of agricultural electricity is used for groundwater pumping. Hence, we have revised section 4.3.3 (pg13/ln8-29) as the following:

A similar analysis has been conducted for food production and groundwater storage change. As the food production has increased there has been a decline in the storage of groundwater (Fig. 6a) and they have a high negative correlation (R= -0.73) as seen in the scatter plot (Fig.6b). On the other hand, a positive correlation (R = 0.96) exists between the food production and electricity consumption (Fig. 6d). This implies the high dependency of food production and groundwater irrigation.

Energy usage in agriculture has two major usages: pumping groundwater for irrigation (electricity) (Scott and Shah, 2004; Birner et al., 2007; Kumar et al.,2013) and mechanisation (diesel) due to the use of tractors (Jha et al., 2012). Several studies have stated that the electricity for agricultural purposes is mainly used for irrigation (Scott and Shah, 2004; Birner et al., 2007; Kumar et al.,2013) because the farm mechanisation is dependent upon diesel (Jha et al., 2012). Hence, an assumption that has been considered for this analysis is that, the data for 'consumption of electricity for agricultural purposes' represent the electricity used for pumping groundwater.

In Figure 6c we observe that in the year 1957-58 the electricity consumption was 544.64 GWh along with a food production of 64.31 million tonnes. 20 years later the electricity consumption increased by nearly 20 times (10107.36 GWh) but the food production increased only by twice the previous (126.41 million tonnes). In year 1997-98 electricity consumption was 97195 GWh but still the food production lagged (203 million tonnes). In the recent year of 2011-12 food production increased to 259.32 million tonnes and electricity consumption was 140960 million tonnes. Hence, over the past 54 years (1957-58 to 2011-12) electricity consumption has increased by more than 250 folds whereas the food production has increased only by four folds. Thus, when actual observed values are considered we see that electricity consumption has increased in leaps and bounds but food production has failed to do so, which brings out the concern regarding food security. Thus, there exists a clear dis-balance between the three facets of the nexus. This indicates that the food security has been maintained at the cost of water and energy security.

• There is a positive feedback between groundwater depletion and electricity consumption for pumping which is hinted at but not conclusively illustrated. This implies a progressive relationship between the groundwater table depth and electricity consumption for pumping. If the data allow demonstrating this, it would be a valuable addition to this research.

We thank the reviewer for stating this important finding and we will include the following analysis in the revised manuscript in section 4.3.2 (pg12/ln13-23).

A similar analysis has been conducted for the progressive relationship between the observed groundwater table depth and electricity consumption for pumping, i.e., there exist a correlation between the annual change in groundwater level (year 2-year1) and the electricity used for pumping in year 2. We plot the same for the three regions, Middle Ganga Basin (MGB, Fig. R2(a)), North-West India (NWI, Fig. R2(b)) and West Central India (WCI, Fig. R21(c)). The duration of the analysis is from 1999 to 2011.

We find statistically significant positive correlation between groundwater level drop and electricity consumption for MGB. However, statistically significant correlation does not exist for NWI and WCI. For WCI, this is expected and it is consistent with overall increase in groundwater level that possibly attributes to judicious use of groundwater. However, a careful investigation for NWI reveals that the correlation value is dominated by two outliers (marked in red in Fig.R2(b)) of changes in ground water table depth. After removing the outliers, we obtain a very high statistically significant correlation as presented in Fig. R3.

We would also like to mention here, that the well data available from CGWB are not continuous (available for only months of Jan, May, August and November) and the sample size is also low. Under such situation, with spatially and temporally discontinuous ground observations, a high correlation may not be expected.

[Figure]

**Fig. R5.** Scatter plots showing the correlation between observed groundwater table depth and agricultural electricty consumption for the three sub region (a) MGB, (b)NWI and (c)WCI.

[Figure]

**Fig. R6.** Scatter plot of NWI after removing the outliers.

• The manuscript is not easy to read because of long composite sentences, poor language (omission of words,
20   contaminations, singular/plural correspondence errors, split composite words, word placement), and unclear references. It would be correct and kind to the reader to correct all of these. I suggest having the final version proofread.

We thank the reviewer for mentioning the above discrepancy in the manuscript, we will revise the entire manuscript with reframed short sentences and better language. References have been added accordingly and the
25   final version has also been proof-read.

• URLs in the body should go into footnotes if allowed or in the references section.
We have taken care of this, reformated the references including the URLs.
• I suggest the use of vector graphics where possible.
30   All the figures have been plotted as vector graphics (.eps) in the revised manuscript.
• It would be good to define food security, water security and energy security.

The following definitions would be added in Sect.1 (pg1/ln10-17) for a better understanding of their interlink

The United Nations has defined food security and water security as the following:

"Food Security is the condition in which all people, at all times, have physical, social and economic access to sufficient safe and nutritious food that meets their dietary needs and food preferences for an active and healthy life."

"Water security is defined as the capacity of a population to safeguard sustainable access to adequate quantities of acceptable quality water for sustaining livelihoods, human well-being, and socio-economic development, for ensuring protection against water-borne pollution and water-related disasters, and for preserving ecosystems in a climate of peace and political stability."

The International Energy Agency (IEA) defines energy security as "the uninterrupted availability of energy sources at an affordable price".

• The summary and conclusions section must not contain new information but it does.

We have edited this section and summarised our work in comparison to other works done, no new information has been provided.

• The references are not formatted consistently, e.g. 'Pai D.S.', 'Panda, D . K.' and 'Pande S' are consecutive and all different.

These inconsistencies will be rectified as the following and all other references will be cross-checked.
Pai, D.S., Sridhar, L., Rajeevan, M., Sreejith, O.P., Satbhai, N.S. and Mukhopadhyay, B.: Development of a new high spatial resolution (0.25° X 0.25°) long period (1901-2010) daily gridded rainfall data set over India and its comparison with existing data sets over the region, Mausam, 65(1), 1-18, 2014.

Panda, D. K. and Wahr, J.: Spatiotemporal evolution of water storage changes in India from the updated GRACE-derived gravity records. Water Resour. Res., 52(1), 135-149, doi:10.1002/2015WR017797, 2016.

Pande, S. and Savenije, H.H.: A sociohydrological model for smallholder farmers in Maharashtra, India, Water Resour. Res., 52(3), 1923-1947, doi:10.1002/2015WR017841, 2016.

• Line-by-line comments:
– 2/16: 'water used as hydro electricity to generate power' does not make sense. Rather say e.g. 'water used for hydropower to generate electricity'
This has been rectified to (pg2/ln22-23):
"Water is required for agricultural produce, energy is required to pump the water from various sources, and again water is used for hydropower to generate electricity."

– 4/9: 'depletion of ground water table' - 'falling groundwater table' or just 'depletion of groundwater'

This has been changed to 'depletion of groundwater' (pg4/ln5).
– 8/16: rate of population growth
This has been rectified(pg8/ln33-34):
"but a steady increase in the rate of population growth."
– 9/1: something is wrong with figures 2g and 2h: the 50000-130000 scale corresponds to electricity consumption in 2g and to expenditure in 2h.

This has been rectified, we apologize for the inadvert error in the axis values.

– 9/10-11: 'zero is the surface level denoting no change in the water table':this confused me - how does the surface level affect changes in the water table?

Here we meant that the groundwater level is at the surface (no fall in the level). We have modified the sentence for better understanding as (pg9/ln27-30):

"Figure 3b, d and f show the groundwater depths measured in meters below ground level (mbgl), where zero refers to the groundwater at the surface (opening of the observation well) denoting no change in the water table. A time series of the four months have been plotted from 1996 to 2014."

– 15/31-32: this claim is incorrect. The study involves but does not encompass all three sectors of water, food and energy. This study refers only to fractions of the energy and water sectors.

We agree with the reviewer but our claim is more concentrated with respect to the agricultural groundwater use and agricultural energy use. We have modified the sentence accordingly as (pg17/ln14-15):

"This present study is the first of its kind for India, encompassing all the three major sectors of water, food and energy from the perspective of the agricultural sector."

**3 Technical corrections**

We thank reviewer 3 for taking time out to point out these intricate errors. Spelling and grammatical errors has been corrected to the best of our knowledge.

The various spelling and grammatical errors are not listed here.

The listed-out errors have been addressed in the revised manuscript as the following:

• 5/7-10: inconsistent formatting of numbers, and decimals completely superfluous.

This has been rectified in the revised manuscript as:

"The three sub-regions studied here are, North-West India (NWI) (the states of Rajasthan, Punjab, Haryana and Delhi) covering an area of 437,739.14 km$^2$; Middle-Ganga Basin (MGB) (the states of Uttar Pradesh and Bihar) covering an area of 339,488.09 km$^2$ and West-Central India (WCI) (the states of Maharashtra and Goa) encompassing an area of 311,249.34 km$^2$."

• general: 'well depth'

This has been rectified in the revised manuscript as suggested.

• 15/16: 'Validation of satellite derived groundwater' - add 'data' at the end of this

This has been rectified in the revised manuscript.

• Fig. 3a: coordinates right and bottom are duplicate and unnecessary.

The duplicate coordinates have been removed.

• Fig. 4a: 'Deficit/Excess' first in legend but on secondary axis (on the right), this is counterintuitive.

This has been rectified by making deficit/excess as the second entry in the legend in Fig. 4a, also the same correction has been made in the supplementary material for Supplementary Fig. S5a, Fig. S8a and Fig. S11a.

[revised manuscript text omitted]